



# Design and operation of a long-term monitoring system for spectral electrical impedance tomography (sEIT)

Maximilian Weigand[1], Egon Zimmermann[2], Valentin Michels[1], Johan Alexander Huisman[3], and
Andreas Kemna[1]

[1]Geophysics Section, Institute of Geosciences, University of Bonn, Bonn, Germany
[2]Electronic Systems (ZEA 2), Central Institute for Engineering, Electronics, and Analytics, Forschungszentrum Jülich GmbH,
Germany
[3]Agrosphere (IBG 3), Institute of Bio- and Geosciences, Forschungszentrum Jülich GmbH, Germany

**Correspondence:** Maximilian Weigand (mweigand@geo.uni-bonn.de)

**Abstract.** Spectral electrical impedance tomography (sEIT) is increasingly used to characterize the structure of subsurface systems. Additionally, petrophysical and biogeophysical processes are characterized and monitored using sEIT. The method combines multiple, spatially distributed, spectroscopic measurements with tomographic inversion algorithms to obtain images of the complex electrical resistivity distribution in the subsurface at various frequencies. Spectral data, as well as polarization

measurements provide additional information about the systems under investigation, and can be used to reduce ambiguities that occur if only the in-phase resistivity values are analysed. However, spectral impedance measurements are very sensitive to details of the measurement setup, as well as external noise and error components. Despite promising technical progress in improving measurement quality, as well as progress in the static characterisation and understanding of electrical polarisation signatures of the subsurface, long-term monitoring attempts are still rare. Yet, measurement targets often show inherent non-

stationarity that would require such approaches for a proper system characterisation. With the aim of improving operating foundations for similar endeavours, we here report on the design and field deployment of a permanently installed monitoring system for sEIT data. The specific aim of this monitoring installation is the characterisation of crop root evolution over a full growing season, requiring multiple measurements per day over multiple months to capture relevant system dynamics. In this contribution, we discuss the general layout and design of the monitoring system, including the core measurement system,

additional on-site equipment, required corrections to improve data quality for high frequencies, data management, and remote processing facilities used to analyse the generated data. The choice and installation of electrodes, cables, and measurement configurations are discussed, as well as quality parameters used for the continuous assessment of system functioning and data quality. Exemplary analysis results of the first season of operation highlight the importance of continuous quality control. It is also found that proper cable elevation decreased capacitive leakage currents and in combination with the correction of inductive

effects lead to consistent tomographic results up to 1 kHz measurement frequency. Overall, the successful operation of an sEIT monitoring system over multiple months with multiple daily tomographic measurements was achieved.



# 1 Introduction

Among non- or minimally invasive methods to characterise the subsurface, electrical methods have the advantage of being relatively inexpensive and easy to use, applicable both at the laboratory and the field scale, and sensitive to a wide variety of lithological, petrophysical, and biological properties and processes (e.g., Kemna et al., 2012; Kessouri et al., 2019; Cimpoiaşu et al., 2020, and references therein). Building upon the established electrical resistivity tomography (ERT) method, spectral electrical impedance tomography (sEIT) captures the subsurface distribution of the frequency-dependent complex electrical conductivity in the mHz to kHz frequency range. The complex electrical conductivity describes both in-phase conduction and out-of-phase polarisation, which are two independent material properties. As such, sEIT combines the imaging capabilities of ERT with the improved information content of spectral induced polarisation (SIP), which is sometimes also referred to as electrical impedance spectroscopy (EIS).

Multiple sources of polarisation signatures have been identified for frequencies below 1 MHz, including polarisation of the electrical double layer (EDL), membrane polarisation, and Maxwell-Wagner polarisation (e.g., Kemna et al., 2012, and references therein). Central to all these sources of polarisation is the movement and accumulation of charge carriers in external electric fields, leading to relaxation phenomena characterised by phase shifts between excitation and response signals. Depending on the subject of investigation, considering the polarisation at different frequencies (i.e., spectral measurements) can enhance the information content with respect to certain physical properties or processes.

sEIT instrumentation has seen a steady development in the past decade (e.g., Zimmermann et al., 2008a, b; Xi et al., 2015; He et al., 2016; Radic, 2016, 2021). Development has shifted from the design of the systems themselves to data error corrections that improve measurement quality, including accounting for capacitive leakage currents and associated electrical pathways and the correction of inductive high-frequency effects (e.g., Zhao et al., 2013; Huisman et al., 2015; Zimmermann et al., 2019; Wang et al., 2020b). The nature of capacitive effects is now better understood, which reduces measurement errors in EIT measurements (Schmutz et al., 2014; Flores Orozco et al., 2021).

Several studies have shown the principal applicability of sEIT at the field scale (e.g., Kemna, 2000; Flores Orozco et al., 2012a; Günther and Martin, 2016; Maurya et al., 2018; Mudler et al., 2021). Yet, most of these studies only analysed frequencies well below 1 kHz. In addition, multiple monitoring studies have been reported. Here, most studies involved larger time spans between measurements (days to weeks), (e.g., Williams et al., 2009; Commer et al., 2011; Flores Orozco et al., 2011, 2013), although Kelter et al. (2018) conducted a short-term tracer monitoring experiment with smaller measurement intervals. As pointed out by Kelter et al. (2018), reliably capturing sEIT data in the field is still challenging and time-consuming. Yet, the potential of fixed monitoring setups with high temporal resolution and a wide spectral bandwidth is huge, as these two characteristics directly control which processes can be captured and differentiated. The situation of long-term sEIT monitoring is similar to that of long-term resistivity monitoring. Such monitoring applications were recently reviewed by Slater and Binley (2021), who argue that fixed measurement installations have a high potential for a multitude of disciplines, yet remain underused and underdeveloped.





With the aim of progressing the state of the art for long-term sEIT monitoring, we present the installation and operation details of a permanently installed sEIT field setup. The scientific motivation for our sEIT monitoring is rooted in the field of biogeophysics, where a high sensitivity of electrical signatures to biogeoechemical structures and processes has been identified (e.g., Atekwana and Slater, 2009; Kessouri et al., 2019). Based on previous laboratory findings that showed the principal capability of sEIT to directly image crop roots (Weigand and Kemna, 2017, 2019), we anticipate that the spatio-temporal

evolution of crop roots under field conditions can be characterised if updated measurement and processing procedures that yield reliable high-bandwidth spectral polarisation monitoring information are implemented.

The remaining text is structured as follows. The next section will shortly introduce procedures for multi-dimensional impedance measurements, after which the experimental setup is presented. The sEIT monitoring system is then described in detail, including processing facilities and procedures. In addition, a novel inductive error correction procedure for complex

cable arrangements is presented and applied to the measured sEIT data. After this, exemplary results of the 2018 measurement season are shown to highlight the importance of using quality control parameters for continuous supervision and assessment of such monitoring systems. In addition, the spectral and temporal consistency of the tomographic inversion results will be illustrated. Finally, the overall system will be discussed in light of potential scientific use, and future avenues for development will be presented.

**2   Multi-dimensional impedance measurements**

EIT measurements are commonly conducted using a four-electrode spread with two electrodes used for current injection and the other two electrodes used to measure a potential difference in the ensuing electrical potential distribution in the subsurface (e.g., Telford et al., 1990; Everett, 2013). The four-electrode spread is used to reduce effects caused by the interface resistances between electrodes and soil (i.e. the contact resistances). Measurements can be conducted in the time- or the frequency-domain,

where time-domain measurements use rectangular wave forms for resistivity and time-domain induced polarisation (TDIP) measurements, while sinusoidal wave forms are used for EIS/SIP or sEIT measurements in the frequency domain. The present study analyses frequency-domain sEIT measurements in the frequency range between 0.1 Hz and 1 kHz. Traditionally, a large number of differently located four-point measurements is analysed by employing error-weighted gradient-based tomographic inversion algorithms, producing 2D or 3D maps of the subsurface (e.g., LaBrecque et al., 1996; Kemna, 2000; Binley and

Kemna, 2005; Rücker et al., 2017).

Tomographic analysis requires careful data assessment as the underlying inversion algorithms assume uncorrelated, normally distributed noise, and any outliers or highly correlated noise components should be removed or at least reduced as much as possible. Correspondingly, data error models have been developed both for ERT (e.g., LaBrecque et al., 1996; Koestel et al., 2008) and for sEIT (Flores Orozco et al., 2012b), which typically rely on the analysis of normal and reciprocal measurements. Nor-

mal and reciprocal measurements hereby refer to measurements where current and voltage dipoles are switched. Theoretically, such a switch should yield the same impedance measurement, and remaining differences can thus be used to quantify random measurement errors. Yet, this approach requires comparable signal-to-noise ratios and small levels of systematic variations in



both normal and reciprocal measurements, and therefore cannot always be used. In this case, heuristics must be employed to find suitable data error estimates for the inversion.

In general, data noise and systematic errors should be reduced and prevented as much as possible. In cases where large amounts of noise are expected, often the so-called 'robust' L1-norm inversion is used (e.g., LaBrecque and Ward, 1990; Kemna, 2000) to reduce the effect of outliers on the inversion results. Although this may provide qualitatively robust inversion results, this strategy does not allow a precise error weighting of the measured impedances and thus is not suitable for quantitative analysis. When quantitative and robust phase polarisation values are to be recovered from noisy measurement environments,

proper data quality control and filtering are the only options. However, integrated data quality control schemes remain rare, with only few examples known for TDIP (e.g., Mitchell and Oldenburg, 2016; Flores Orozco et al., 2018) and for FDIP (e.g., Flores Orozco et al., 2013; Katona et al., 2021).

Inversions presented in this study were conducted using CRTomo (Kemna, 2000) following a two-step approach. First, a fully complex inversion is implemented minimizing the following objective function:

$$\Psi = \Psi_d + \lambda \Psi_m = \left|\left|\hat{\mathbf{W}}_m(\hat{\mathbf{d}} - \hat{\mathbf{f}}(\hat{\mathbf{m}}))\right|\right|_{\mathrm{L2}}^2 + \lambda \left|\left|\mathbf{R}_m\hat{\mathbf{m}}\right|\right|_{\mathrm{L2}}^2, \tag{1}$$

where $\Psi_d$ is the data misfit term consisting of the error-weighted residuals and $\Psi_m$ is the model objective function providing stabilisation and regularisation to the inversion problem in the form of a model smoothness constraint. The complex data vector $\hat{\mathbf{d}}$ consists of logarithmically transformed complex transfer conductances, $\hat{\mathbf{f}}$ is the forward response for a given model vector $\hat{\mathbf{m}}$ of log-transformed complex conductivities, and $\hat{\mathbf{W}}_m$ is the complex diagonal data weighting matrix containing

the reciprocal complex data error estimates $1/\hat{\sigma}_i$ on the main diagonal. $\mathbf{R}_m$ is the regularisation matrix that imposes a 2D smoothness constraint on the model parameters. It should be noted that the use of log-transformed complex conductivities implies an inversion of log-magnitudes and linear phase values. In a second step, a real-valued, phase-only inversion with fixed conductivity magnitude values, the so-called final phase improvement (FPI), is used to improve the estimates for the phase (details on this procedure can be found in Kemna, 2000). Note that the interface for the inversion code is formulated

in complex impedances, i.e. the inverse of the complex conductances, and therefore the data errors are discussed in terms of impedance values later on.

Data inversions were performed independently for each frequency and point in time, as was done in other studies (e.g., Kemna, 2000). It is however possible to impose additional constraints along the time (e.g., Karaoulis et al., 2013) or frequency axis (e.g., Commer et al., 2011; Kemna et al., 2014; Günther and Martin, 2016) to smoothen noise components between

neighbouring frequencies or time steps.

## 3  The Selhausen rhizotron facility

The sEIT monitoring system described in this study is part of the Selhausen rhizotron facility described in detail in Cai et al. (2016). The facility consists of an agricultural test field of 9.75 m width and 7 m length with a basement running along the width down to a depth of 2.25 m at one side of the field (Fig. 1). Various sensors have been installed on this side of the field for





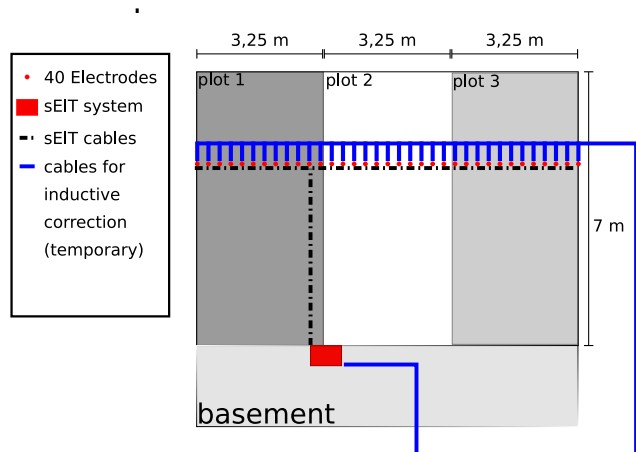

**Figure 1.** Overview of the field site, including location of measurement system (in a basement next to the measurement field), the cable layout of the sEIT system, electrode positions, and calibration cable layout temporarily used for measurement of inductive calibration data (discussed in section 4.3).

hydrological and biological monitoring. The installed sensors include temperature sensors and Time-Domain Reflectometry (TDR) probes for water content measurements at different positions along the profile and at different depths, as well as optical rhizotubes with 7 m length used for optical root observations (not further discussed here, see Cai et al. (2016) for further details). The facility is well suited for an sEIT monitoring system aimed at biogeophysical characterisation of agricultural fields because of the availability of temperature data for data correction, TDR-derived water content data at certain locations
to support data interpretation, as well as a mains power and high-bandwidth internet connection.

## 4 Design of the monitoring system

The sEIT monitoring system is comprised of a 40-electrode profile with 25 cm electrode spacing (Fig. 1). The system consists of two interconnected parts: an on-site cluster of hardware used for the actual sEIT measurements and data storage, and a remote computer network used for the final data archiving and processing (Fig. 2). The on-site part is comprised of the actual
sEIT measurement system with attached electrodes, and two minicomputers used as a GPS (Global Positioning System)-based time server and data server, respectively. System times are set to UTC to prevent any time conversion issues. The on-site system is connected to the remote site using two internet connections: a broadband internet link provided on-site, and an additional connection through a mobile wireless-broadband connected VPN (virtual-private network) network. This redundancy in connections was required to deal with temporary malfunctioning of the connection equipment. The remote
site consists of a RAID (Redundant Array of Independent Disks)-6 protected file server and associated independent backup solutions, a Postgresql database server, and multiple multi-core computers controlled by a task queuing system for computing-intensive tasks. While remote supervision is an integral part of the system, it can also be operated semi-autonomously in case





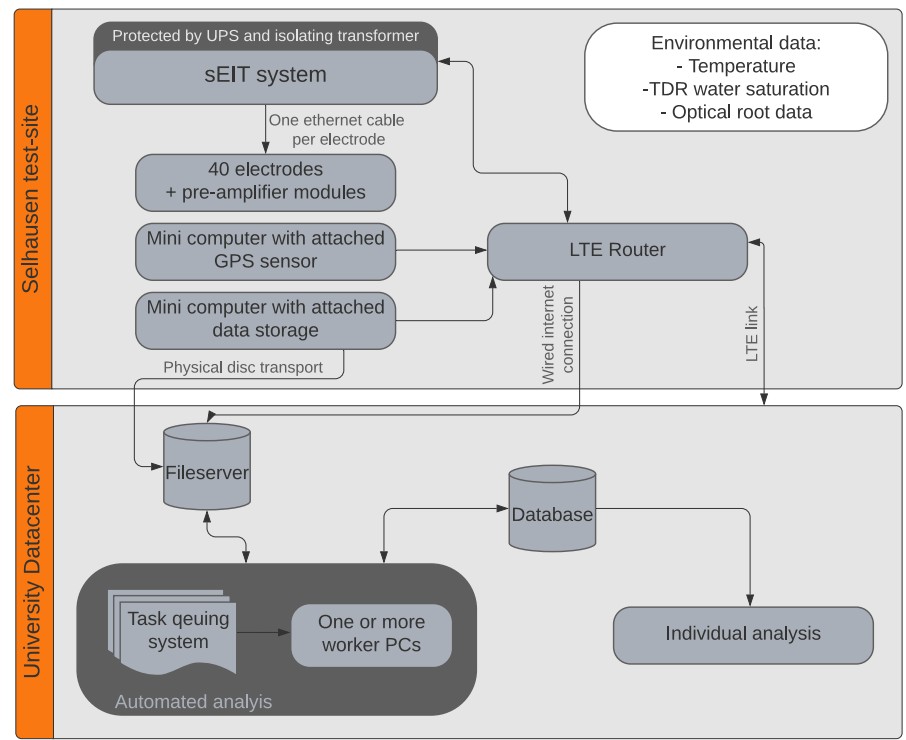

**Figure 2.** Schematic overview of sEIT monitoring system. The system is divided into two major parts, an on-site part (top box) and a remote part (bottom box).

of connection loss. The system has sufficient data storage and backup capacity to operate independently for a few weeks, which allows for flexible scheduling of maintenance visits.

### 4.1 sEIT measurement system

The core of the sEIT measurement system is described in Zimmermann et al. (2008b) and Zimmermann et al. (2019). It is a 40-channel EIT system that measures the electrical impedance in the frequency range between 1 mHz and 45 kHz. For each current injection dipole, time-series of the potentials of all remaining 38 electrodes with regard to system ground are measured. For this, each electrode is equipped with a small pre-amplifier module to reduce cable effects and increase input impedances to 500 GΩ. Subsequently, the potential signal at the excitation frequency is extracted using a digital lock-in approach, which results in three-point transfer impedances for each current injection and each potential electrode. From these 'three-point' measurements, arbitrary four-point configurations are computed by superposition for all of the specified current injection dipoles. The potential measurements relative to system ground require the isolation of the system from the power grid by means of an isolating transformer to prevent ground loops. A symmetric voltage of $\pm 9$ V (18 Vpp) is used for current injection, and two shunt-resistances of 1 kΩ are used for current measurement at each of the two current electrodes. While the system





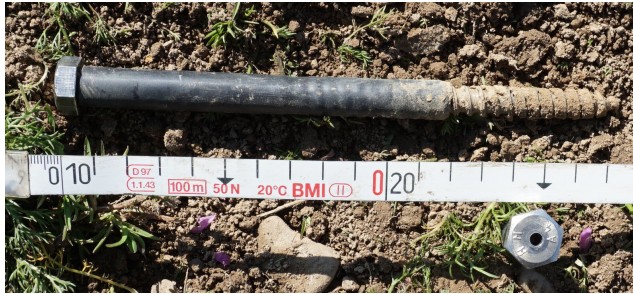

**Figure 3.** Stainless-steel electrode used for sEIT measurements. Scale is in cm.

could in theory support an excitation voltage of 10 V, a lower excitation voltage was chosen to prevent ADC (analog-to-digital converter) saturation. In order to enable autonomous measurements, the sEIT system was modified to enable daily calibration measurements (discussed in section 4.3) without any user interaction and physical access to the system.

Due to the specific position in the current path, the shunt resistances for each current channel form voltage dividers with respect to the subsurface (dominated by the contact resistances), effectively distributing the applied voltage on shunt resistors and the subsurface in a series circuit. Therefore, the shunt resistances need to be roughly adjusted to the encountered contact resistances, as discussed in Zimmermann (2011). If the selected shunt resistances are too small, the measured voltage drops at the shunt resistors will be small, thereby reducing the precision of the current measurements. If the shunt resistances are too large, this will lead to unnecessarily large voltage drops over the shunt resistors, and more importantly a reduced voltage

actually applied to the subsurface. The choice of 1 k$\Omega$ for the shunt resistors turned out to be an appropriate choice for the measurement time span discussed in this study, since it matched well with the encountered range of contact resistances.

### 4.2 Design of electrode and cable layout

Medical-grade V4 steel screws were used as electrodes for both current injection and potential measurements (Fig. 3). While the use of non-polarisable electrodes would have been preferred, no sufficient solution for long-term measurements was found

and steel electrodes were deemed a good compromise between long-term stability and the ability to measure polarisation signatures. Long-term suitability entails non-leakage of electrode fluids, which, for example, is not the case for $Cu/CuSO_4$ unpolarisable electrodes sometimes used for SIP/sEIT measurements. It also requires sufficient contact with the soil even in dry conditions, and stable internal electrical characteristics. The steel screws are 18 cm long with a diameter of 1 cm. The area of current injection was limited to the lower 5 cm of the electrode by covering the upper 12 cm of each screw (plus 1 cm for

screw head) with two layers of shrinking tube (Fig. 3). By only exposing 5 cm of the electrode to the soil, we aimed to stay as close as possible to the point-electrode assumption used in modelling and inversion, while ensuring sufficient soil contact to keep the contact resistance in an acceptable range also in dry conditions.

In the first monitoring period (growing season 2017), electrodes were driven 5 cm into the soil to fully connect the bare part of the electrode with the subsurface (Fig. 4, left). The connecting cables were bundled with cable ties and routed to the





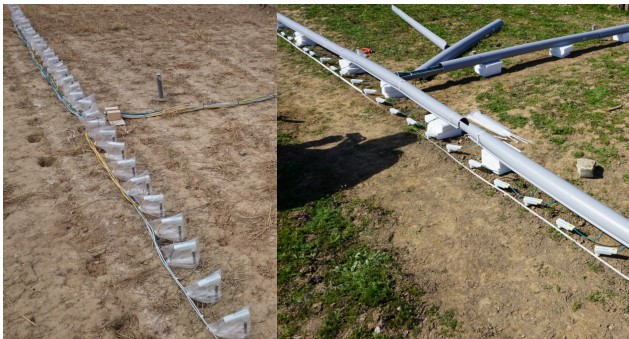

**Figure 4.** Installation of electrodes, electrode modules, and connecting cables. Left: Installation in 2017, right: Installation in 2018.

measurement system as a cable bundle lying on the soil surface. In the second monitoring period (growing season 2018), the
setup was modified and electrodes were driven 15 cm into the ground, with only 3 cm remaining above the surface. As such,
the bare part of the electrode started at a depth of 10 cm and reached down to 15 cm depth. Cables were bundled and sheltered
against the environment in rain gutters, which were elevated from the ground using Styropor blocks with a thickness of 10 cm
(Fig. 4, right). Excess cable lengths were arranged in elongated loops parallel to the cables within the rain gutters to ensure that
the electromagnetic induction effects of these excess cable lengths cancelled each other out.

### 4.3   Calibration and correction measurements

Based on previous work, sEIT measurements were corrected daily for system drift, variable ADC channel gains, signal prop-
agation, and current channel capacitances, as presented and discussed in Zimmermann et al. (2008b); Zimmermann (2011);
Zimmermann et al. (2019). In addition, inductive cable-to-cable effects were corrected using a a modified version of the correc-
tion procedure previously developed for borehole (Zhao et al., 2013, 2015) and surface EIT applications (Zimmermann et al.,
2019; Wang et al., 2020b), which will be presented in detail below. Capacitances between all cables and system ground were
also measured daily to assess the potential for leakage currents and as an indicator of system health (e.g., Zimmermann et al.,
2019). While these capacitances were not used for the correction of data in this study, they can potentially be integrated into
forward modelling algorithms to account for these additional current pathways.

### 4.3.1   Correction of inductive effects

Inductive effects on electrical impedance measurements are two-fold. First, mutual impedances between the cables of the
measurement system induce corresponding voltages directly on the potential measurement cables. Second, currents induced in
the soil by inductive effects lead to a change of the earth response measured at the potential electrodes. We here deal only with
the first effect, as this effect is strongest for the short electrode lay-outs used in this study. A short discussion on the strength of
the earth response can be found in Zhao et al. (2015) and Wang et al. (2020b).


Following Zhao et al. (2015), the effect of the mutual cable inductance $L$ appears as a purely imaginary component $j\omega L$ in the measured complex transfer impedance $\hat{Z}_t$ at angular frequency $\omega$ using current electrodes $A, B$ and potential electrodes $M, N$:

$$\hat{Z}_t^{ABMN} = \frac{\hat{U}_{MN}}{\hat{I}_{AB}} = \hat{Z}_{\text{soil}} + j\omega L_{ABMN}, \tag{2}$$

where $\hat{I}_{AB}$ is the injected current between electrodes A and B, $\hat{U}_{MN}$ is the measured voltage between the electrodes $M$ and $N$, and $j$ is the imaginary number. As the cable inductance acts on the transfer impedance, one direct consequence is that measurement configurations with higher geometric factors are potentially affected stronger by the mutual cable inductance due to the corresponding decrease in transfer resistance. The mutual inductance $L_{ABMN}$ of a four-point spread is computed by superposition of the mutual inductances of all involved cable pairs, accounting for current flow direction by changing signs
accordingly:

$$L_{ABMN} = (L_{AM} - L_{AN}) - (L_{BM} - L_{BN}). \tag{3}$$

For each cable pair $(C1, C2)$, the mutual inductance can be computed by numerically evaluating the Neumann integral along the involved cable paths $\mathbf{S_1}$ and $\mathbf{S_2}$ (e.g., Sunde, 1968; Henke, 2015):

$$L = \frac{\mu}{4\pi} \int_{\mathsf{C1\_start}}^{\mathsf{C1\_end}} \int_{\mathsf{C2\_start}}^{\mathsf{C2\_end}} \frac{d\mathbf{S_1}\,d\mathbf{S_2}}{r} \tag{4}$$

where $\mu$ is the magnetic permeability and $r$ is the distance between any two cable elements. The start position of each cable $i \in (1, 2)$ is denoted by $\mathsf{C}i\mathsf{\_start}$, the end position by $\mathsf{C}i\mathsf{\_end}$. Due to the inverse dependence on $r$, errors in the spatial location of cables affect the results stronger for smaller separations between the cables. For the sEIT setup used here, the exact positions of the individual cables within the cable bundles were not precisely known, which introduces uncertainty in the determination of the mutual inductance with Eq. (4). To improve the determination of the mutual cable inductances, an additional calibration
measurement was conducted to capture all effects of the cable bundles not properly accounted for in the numerical simulations with Eq. (4). The calibration measurements used additional cables for short-circuit measurements and will be described in detail in the next section. Numerical simulations with the original and the calibration setup were then used in conjunction with the experimental results to obtain improved estimates of the mutual inductances of all cable pairs of the measurement setup. If we denote $L_{C_1 C_2}^{\text{num,measured}}$ as the numerically determined mutual inductance between two cables $C_1$ and $C_2$ in the
measurement setup, $L_{C_1 C_2}^{\text{num,calib}}$ as the numerically determined mutual inductance between those cables in the calibration setup, and $L_{C_1 C_2}^{\text{meas,calib}}$ as the experimentally determined mutual inductance between both cables in the calibration setup, we estimate the corrected mutual inductance between two cables using

$$L_{C_1 C_2} = L_{C_1 C_2}^{\text{num,measured}} + (L_{C_1 C_2}^{\text{meas,calib}} - L_{C_1 C_2}^{\text{num,calib}}). \tag{5}$$

The second term in Eq. (5) represents the difference between the measured and modelled mutual inductance for the calibration
setup. It is assumed that this difference is mainly affected by the uncertainty in the cable positioning within the cable bundles, and can thus be used to improve the numerically determined mutual inductance between the cables in the measurement setup.



For easier handling of the mutual inductances, they are summarised in a pole-pole matrix (Zhao et al., 2015) which represents all possible combinations of cables for the 40-cable setup:

$$
L_{40\times40} = \begin{bmatrix}
0 & L_{1,2} & \ldots & L_{1,39} & L_{1,40} \\
L_{2,1} & 0 & \ldots & L_{2,39} & L_{2,40} \\
\vdots & & \ddots & \ldots & \\
L_{39,1} & L_{39,2} & \ldots & 0 & L_{39,40} \\
L_{40,1} & 0 & \ldots & L_{40,39} & 0
\end{bmatrix}.
\tag{6}
$$

This representation simplifies the computation of the mutual inductance of arbitrary four-point configurations using Eq. (3), and retains the full potential of the system with respect to the voltage measurements relative to ground. Self-inductances for the cables are ignored at this point (hence the zero entries on the diagonal) as they are not relevant for the four-point EIT measurements that the pole-pole matrix is used to correct.

### 4.3.2 Calibration measurements for inductive correction

Following Zhao et al. (2015), calibration measurements were made by adding a short-circuit cable that connected all electrodes to the system ground while ensuring that no contact to the soil or the electrode was established (Fig. 1, blue line). The actual measurement setup remained unchanged during this calibration measurement. The cable used for short-circuiting was connected to the system ground while keeping a large separation to the electrodes and the other cables (see Fig. 1, blue line). Therefore, any current injected using one channel could flow back to the system ground through this cable. For the calibration,

current was subsequently injected into all 40 electrode channels, and potentials towards system ground were measured for all remaining 39 channels. These potential measurements captured the inductive effects on each channel due to current injection at all other channels. This set of calibration measurements consists of $40 \cdot 39 = 1560$ individual measurements of the transfer impedance $\hat{Z}_{C_1 C_2}$ for different pairs of cables arranged in a 40x40 pole-pole matrix similar to Eq. (6). Assuming that only inductive effects affect the potential measured at each channel in these calibration measurements, the mutual inductance can

be computed for each electrode (cable) pair using

$$
L_{C_1 C_2}^{\mathrm{meas,calib}} = \mathrm{Re}\left(\frac{\hat{Z}_{C_1 C_2}}{j\,\omega}\right).
\tag{7}
$$

The mutual inductance thus obtained can be used to correct the actual measurements for inductive effects using Eq. (5).

### 4.3.3 Modeling of cable-to-cable induction

The mutual inductances $L_{C_1 C_2}^{\mathrm{num,measured}}$ and $L_{C_1 C_2}^{\mathrm{num,calib}}$ in Eq. (5) were computed by numerically solving the Neumann integral

(Eq. 4). One basic assumption here is that the position of the short-circuit line can be determined with sufficient precision to correctly compute its effects on the measurement setup. Requirements on positions for the short-circuit line are much less strict in comparison to the measurement setup due to the large distance of all short-circuit cable elements to the original measurement


setup. The numerical solution of Eq. (4) only converges if a finite distance between all cable elements is ensured (due to the $1/r$ singularity in the equation). Therefore we chose an arbitrary cable distance of 1 mm for the numerical representation of

cables within the measurement setup. This assumption must not necessarily coincide with reality as it is contained in both $L_{C_1 C_2}^{\mathrm{num,measured}}$ and $L_{C_1 C_2}^{\mathrm{num,calib}}$, and therefore any difference with the actual separation is expected to be corrected using Eq. (5). Finally, the self inductance of the short-circuit line must be accounted for when solving the Neumann integral, as this part of the calibration setup is connected to both cables $C_1$ and $C_2$. This was realised by numerically replacing the short-circuit line with two lines with a distance equal to the cable radius, as discussed in Henke (2015).

### 4.4 Measurement configurations

The choice of measurement configurations (i.e., the set of four-point spreads) used for tomographic analysis is known to have a large impact on the resulting image resolution and data quality (e.g., Loke et al., 2014, and references therein). While in theory a complete set of measurements could be used to generate all other possible measurement configurations by means of superposition (Xu and Noel, 1993), this approach generally is unfeasible due to the corresponding propagation of errors and the

required removal of data points due to systematic errors. Therefore, heuristically determined measurement configurations were used. Due to the small excitation voltage of 18 Vpp (from -9 to +9 V), large dipoles skipping 20 or 21 electrodes, respectively, were used for current injection. This corresponds to 5.25 m or 5.5 m distance between current electrodes. This resulted in relatively large signal-to-noise ratios in the subsequent potential measurements. A classical dipole-dipole scheme was rejected after test measurements yielded inconsistent data even for small dipole separations, possibly due to small signal-to-noise ratios

for configurations with large geometric factors.

Given that the EIT system used here measures electrode potentials with respect to system ground, only current injections need to be specified before data acquisition. Afterwards, arbitrary four-point spreads are computed using superposition of three-point measurements, which is only limited by the available current injection dipoles. Here, the potential dipoles were selected from the group of all current dipoles and by using gradient dipoles located between the current electrodes (only dipoles with at

least 15 electrodes between M and N electrodes).

### 4.5 Operating schedule and data flow

The actual operation of the sEIT measurements is controlled by a Python-based scheduling program running on the controller of the sEIT device. Therefore, operation does not rely on any additional components in the local or remote network. Daily measurements begin by conducting one ADC calibration measurement which is used to correct for amplification errors in the

system. A subsequent calibration measurement of the capacitive load of the current paths is used to better estimate the true injected currents. These calibration measurements are followed by one measurement of cable capacitances used to estimate the susceptibility of the system for leakage currents. After these daily calibration measurements, tomographic sEIT measurements commence at the beginning of each full hour. With an approximate measurement time of 1.5 h, this results in one tomographic measurement every two hours. A time window is reserved for data transfer at 12 AM each day. The length of this transfer time

window is adjusted to the available internet connection speed.





Data is automatically transferred to a disk attached to a local network using a Raspberry Pi mini computer, from where essential data are then transferred to the remote system component using the wireless broadband VPN connection. Here, all measurement data are stored in their original form on a suitably protected disk server (with back-up functionality). Every 6 hours, essential data is imported into a SQL database (Postgres) with subsequent automated data quality assessment (filtering) and preliminary tomographic analysis.

## 4.6 Data processing

In a first processing step, four-point measurements are processed with regard to the sign of the geometric factor. If the geometric factor is negative for a given configuration, the potential electrodes are swapped to obtain a positive geometric factor. This also ensures that the phase angle of the complex impedance lies in the range between $-\pi$ and $\pi$ radians. This procedure of correcting the sign of the complex impedance values is not strictly necessary for further processing, but allows for easier data analysis (i.e., histogram analysis and setting of filter criteria).

In a second processing step, the data are filtered for outliers, which is essential for tomographic inversion. We attempt to make filtering decisions based on systematic and argument-based criteria, but a large subjective component to filtering remains inevitable. Often, the filtering process is iterative with detailed analysis of individual measurements alternating with checks of the entire data set to ensure that a filter tailored to a specific case does not remove good data at another time step or frequency.

Ultimately, we used two slightly overlapping types of filters in this study. The first type represents filters derived directly from technical considerations based on the system design. In particular, this type of filtering includes the following criteria:

- Contact resistances between current electrodes must be below $40\,\mathrm{k\Omega}$. This ensures that the voltage drop over the shunt resistors remains sufficiently high for an exact current measurement, and measurement errors associated with high contact resistances are kept low.

- Resistances and apparent resistivities must be positive. Negative resistances as well as negative apparent resistivities are physically implausible for surface measurements.

- Cable capacitance for the measurement day must be below 400 nF. This ensures reasonably low leakage currents. Too high values also indicate issues with the system.

The second type of filter relies on the basic assumption of consistency with varying frequency. We are not aware of any processes that lead to a jagged frequency response in EIT measurements and thus we decided to filter out all spectra that are not smoothly varying. Two examples for phase spectra that were deemed acceptable for further use are shown in Fig. **??**a, b, and two examples for filtered phase spectra are shown in Fig. 5 c, d. However, the definition of smooth is clearly somewhat ambiguous and arbitrary. After testing, the following filters that represent different approaches to select suitable spectra were defined (for some filters labels are defined for better reference later on):

- Two frequencies (50 Hz and 110 Hz) were completely removed from the data set due to suspected 50 Hz harmonic noise contamination.



- Impedance phase angles were required to fall between $\phi > -100$ mrad and $\phi < 20$ mrad. These values are based on histogram analyses of selected data, and on our experience of what range of phase values can be expected. Small positive phase values account for the possible occurrence of the negative IP effect (e.g., Wang et al., 2020a).

- (SMOOTHNESS) A smoothness filter was applied based on an $L_1$-norm of the spectral phase variation, and measurements were disregarded when this norm was above an empirically derived value of 3 (with phase values in mrad and frequency value in Hz):

$$L_1^\phi = \sqrt{\frac{1}{N-1} \sum_{i=1}^{N-1} \left| \frac{\phi_{i+1} - \phi_i}{log_{10}f_{i+1} - log_{10}f_i} \right|} < 3, \tag{8}$$

with $N$ the number of data points in each spectrum, $\phi_i$ the corresponding $i$-th phase value of the spectrum, and $f_i$ the $i$-th frequency.

- (SHIFT) A maximum change filter was applied that detects sudden jumps in the phase spectra. Here, the maximum change normalised by the log-distance in frequency was used as a threshold filter (with phase values in mrad and frequency values in Hz):

$$\mathrm{MAX} \left( \sum_{i=1}^{N-1} \left| \frac{\phi_{i+1} - \phi_i}{log_{10}f_{i+1} - log_{10}f_i} \right| \right) < 9.5. \tag{9}$$

The used threshold value of 9.5 was again determined empirically by inspection of the spectra.

- (NR_DATA) After application of all filters, only those spectra were retained which had more than 85 % of the original frequency data points left. This filter was especially important to produce consistent data sets.

The number of data points retained after filtering is shown in Fig. 6 for all time steps of the growing season 2018. It can be seen that the smoothness filter is the only filter that changes its effects significantly over time. Also, we expect some degree of redundancy in the data filters, which has not been investigated in detail. Adding further to this it is important to realise that the large number of applied filters should not necessarily be equated with bad data quality. The nature of EIT systems that measure with respect to the system ground potential allows to obtain a large number of electrode configurations after actual data acquisition, and the filters are here not only used to remove bad, noise-affected data points, but also as a means to select data points for inversion out of the many possible combinations.

Using the 40 current injections specified before the measurements, 1650 four-point measurement configurations were initially generated from the three-point measurements. After applying the various filters to all time-steps only 993 unique measurement configurations were still present in at least one of the filtered data sets analysed for 2018, with the individual number of retained configurations per individual measurement varying (blue area in Fig. (6)). After filtering the 993 remaining configurations had a mean geometric factor of 6.66 m, with a minimum of 0.82 m and a maximum of 135.62 m.

No analysis of normal-reciprocal pairs was performed in this study given that only 40 injections were conducted, thus limiting the maximum number of available normal-reciprocal pairs to this small number. The small number of current injections





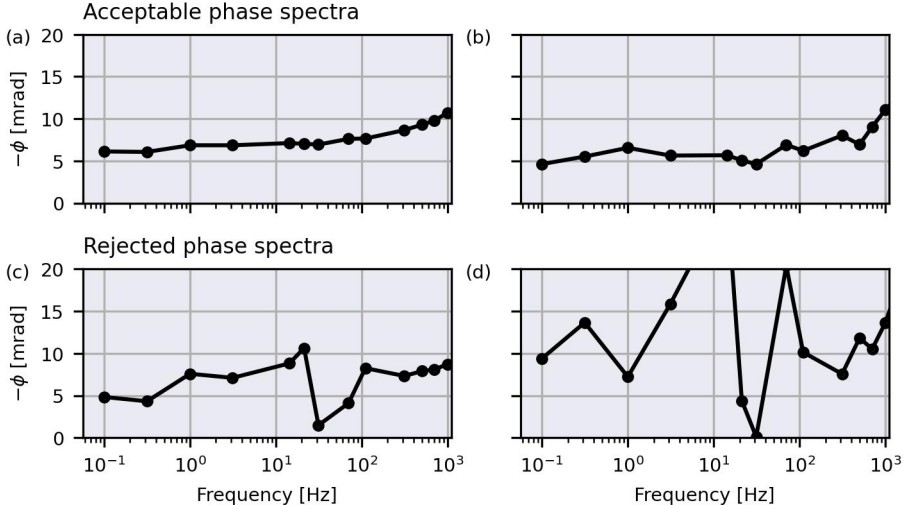

**Figure 5.** a, b: Examples of phase spectra that were deemed acceptable for further use with regard to the smoothness criterion. c, d: Examples of phase spectra that violate the smoothness criterion. Note that the measurements at 50 Hz and 110 Hz were already removed.

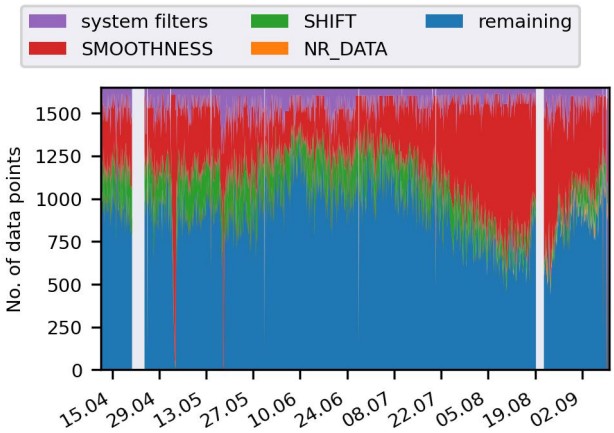

**Figure 6.** Number of data points retained for 1 Hz after application of all filters denoted with corresponding labels in the text. The specific nature of the filters is explained in the text. 'system filters' refers to all system-specific filters.





reduced the required measurement time for individual tomographic data sets and thus allowed for relatively fast acquisition
while still providing an acceptable amount of impedance measurements because of the measurement-to-ground nature of the
350 system.

## 4.7 Data inversion

Inversions were performed for each frequency independently up to 1 kHz. While proper error estimation helps to prevent over-
fitting, in reality it is often difficult to estimate the errors and as such heuristics are commonly employed to define measurement
errors. In this study, the results of the non-linear least-squares inversions were assessed using the normalised root mean square
(RMS) error, which indicates to which degree the measurements are predicted by the inverted model (e.g., Kemna, 2000). It
is common to only consider inversion results with RMS values close to one, indicating a fit of the measurements within their
uncertainty. To find consistent data error estimates, based on a data error model, without the use of normal-reciprocal measure-
ments, a systematic parameter search was conducted. We found that an impedance magnitude error model parameterised as
$\Delta|\hat{Z}| = 0.0005|\hat{Z}| + 0.05\ \Omega$ yielded suitable resistivity magnitude results with RMS values near one for the entire monitoring
period, indicating that an absolute error dominated the resistivity magnitude data. However, no single absolute phase error es-
timate yielded suitable phase RMS values for the entire measurement period. Therefore, a different approach was used. Based
on the assumption that the phase error is dominated by an absolute error value, inversions for all frequencies and all time steps
were conducted for absolute phase error estimates between 0.1 and 1.3 mrad in 0.1 mrad increments. A small relative error of
$0.5\%$ of a given phase value was also added, although its impact on the results was not investigated in detail. In the following,
phase inversion results for each pixel were computed as the median phase value of all inversion results yielding RMS values
between 0.95 and 1.05. Despite limiting the analysed results to appropriate RMS values, sometimes those results still included
artefacts or overly smooth images, and taking the median pixel values attempts to reduce any impact of these unwanted results.

## 4.8 Temperature correction

Complex conductivity distributions obtained through inversion are approximately corrected for temperature variations follow-
370 ing the approach of Hayley et al. (2007). In this approach, the complex conductivity $\hat{\sigma}$ of each inversion cell and frequency is
corrected using:

$$\hat{\sigma}_{\mathrm{corr}} = \left[ \frac{m \cdot (T_{\mathrm{ref}} - 25) + 1}{m \cdot (T - 25) + 1} \right] \cdot \hat{\sigma}, \tag{10}$$

where $\hat{\sigma}_{\mathrm{corr}}$ is the temperature-corrected complex conductivity, $m$ is an parameter set to 0.02 indicating a temperature change
of 2 % per degree Celsius, $T_{\mathrm{ref}} = 19.9°C$ is the reference temperature used for correction, and $T$ is the actual temperature in
$°C$. The reference temperature is chosen as the median temperature observed for the growing season 2018 across all sensors,
and actual temperatures are gathered from temporally and 2D-spatially linearly interpolated soil temperature data from sensors
installed in various horizontal and vertical positions (for more information on the position of the temperature sensors, see Cai
et al., 2016).





Eq. 10 assumes that the temperature acts equally on the real and imaginary part of the electrical conductivity. Several studies
have provided evidence that both the real and imaginary parts show a linear dependence on temperature (e.g., Binley et al.,
2010; Zisser et al., 2010; Bairlein et al., 2016). Yet, these studies also indicated that this linear dependence could depend
on frequency if distinct polarisation peaks are present. Given that no prominent large polarisation peaks were observed in
this study, we assume that the relatively simple correction according to Eq. 10 is sufficient. However, temperature correction
remains challenging especially for field studies and should be addressed in more detail in future studies.

## 5 Results

### 5.1 Operating history

The initial system was installed in April 2017 and monitoring started on 5 May 2017. The monitoring was stopped at 19
September 2018. The growing season 2017 and the following winter were used to test and improve the operation of the
system and to identify relevant factors for obtaining accurate wideband sEIT monitoring data. This test period consisted of
2970 tomographic sEIT measurements. After analysis, it was concluded that these initial data were not of sufficient quality
for reasons explained later. The electrode and cable layout was modified in the beginning of April 2018. After this change
in design, 1769 sEIT measurements were conducted in the growing season 2018 (an average of 11 daily measurements). The
average daily data volume ranged from 5 to 9 Gb, depending on the measurement schedule that was occasionally adapted to
accommodate available upload bandwidth and maintenance times.

### 5.2 Inductive calibration measurements

The measurement setup was retrospectively calibrated for mutual induction effects in the summer of 2020. This was possible
because the setup including the cable lay-out was unchanged since April 2018. The pole-pole matrix obtained from the cal-
ibration measurement is shown in Fig. 7a. Unfortunately, 6 electrode cables were damaged by animals after the monitoring
ended in 2018. Therefore, a spatial linear interpolation of the measured data was used to fill in the missing data to obtain a full
pole-pole matrix of mutual inductances. The measured pole-pole matrix shows that variation in mutual inductance between
cables occurred in groups of eight (Fig. 7a). We attribute this to the fact that the cables were tightly fit together with cable ties
in bundles of eight, thereby decreasing the distance to each other and causing increased mutual inductances. Next, the mutual
inductances were calculated for the calibration set-up (Fig. 7b) and the measurement setup (Fig. 7c) using Eq. 4. In a final step,
the pole-pole matrix used to correct the measured impedances was obtained using Eq. (5) (Fig. 7d). A comparison of Figs. 7a
and 7b shows that the simulation of the calibration setup captures the overall structure of the measured pole-pole matrix well,
but that details are lacking because of insufficient knowledge about the actual positions of the cables.

The effect of the correction for inductive coupling varies strongly between four-point configurations. Fig. 8 presents the
changes in SIP spectra for two configurations associated with the smallest and largest change in the imaginary part of the
transfer impedance, $Z''$, due to the correction for inductive coupling for a selected measurement data. The data set was selected





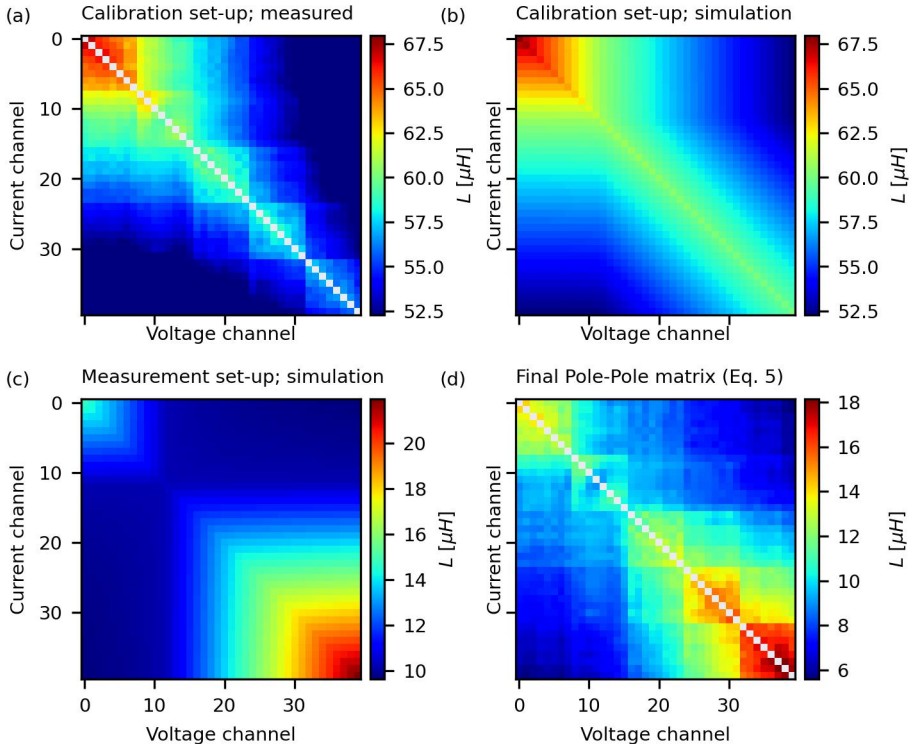

**Figure 7.** (a) Pole-pole matrix of the inductive calibration measurements. Missing electrode rows and columns were filled with a 2D linear interpolation. (b) Result of numerical simulation of calibration setup. (c) Result of numerical simulation of the measurement setup. (d) Final pole-pole matrix computed from calibration measurements and numerical simulations using Eq. (5).

because it showed the smallest mean impedance values (for both real and imaginary parts) and thus the potentially strongest effects of inductive coupling (refer to Eq. 2). The spectrum that is not much influenced by inductive coupling only shows differences above 1 kHz after calibration (Fig. 8a). Yet, there is some potential for misinterpretation of the uncorrected data as the spectrum shows a peak at 2 kHz that is removed by the inductive correction procedure. The second spectrum shows physically implausible behaviour at higher frequencies above 100 Hz with a sign change of $Z''$ (Fig. 8b). The inductive correction corrects this behaviour, thereby increasing the usable frequency range into the kHz region.

## 5.3 Continuous assessment of data quality

Contact resistances for each current injection were automatically determined, allowing a detailed assessment of the temporal dynamics of the contact resistances. The mean contact resistance for all current injections along with the minimum and maximum values are presented in Fig. 9. The contact resistances showed a strong temporal and spatial variation in summer 2017, and stabilised with the beginning of the winter season with only a gradual increase until February 2018. After the redesign

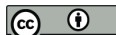

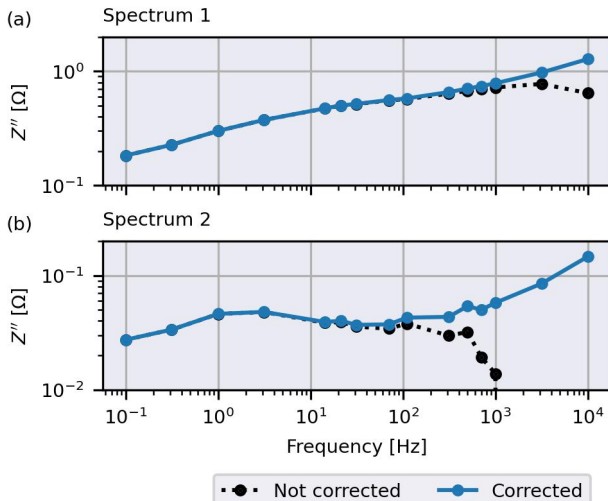

**Figure 8.** Effect of correction of inductive coupling for two spectra of imaginary parts of the transfer resistance, $Z''$. (a) Least affected spectrum measured on 16 June 2018 (11 AM UTC) (b) Most affected spectrum measured on 16 June 2018 (11 AM UTC).

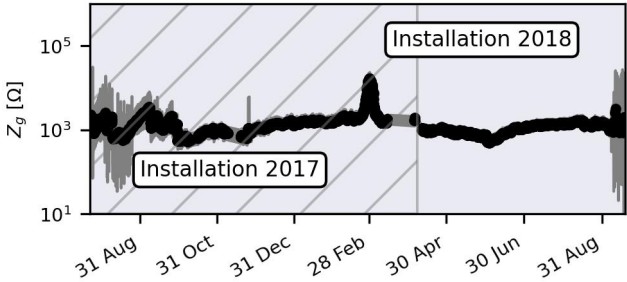

**Figure 9.** Evolution of mean contact resistances for all current injections at 1 kHz. Gray areas indicate min and max ranges.

of the electrode and cable setup in March 2018, the contact resistances showed much less variation until a sharp increase in variation was observed in late summer again.

Contact resistances are dominated by the interface resistance between electrode and soil, and changes in soil properties are typically not considered to be a major factor. The observed behaviour of the contact resistances thus suggests that the electrode

contact to the soil varied considerably in 2017, probably due to the shallow penetration depth. The upper few cm of soil are subjected to cycles of wetting and drying, which may lead to swelling and shrinking and other mechanical changes of the top layer. During the redesign, the active part of the electrode was inserted deeper into the soil. This strongly reduced the variability in the soil-electrode contact, probably due to reduced variation in soil water content and temperature as well as more homogeneous soil structure. Finally, at the end of the growing season of 2018 the contact resistances increased again.

Analysis of only the contact resistances suggests that this increase is associated with increasing loss of contact with the soil due



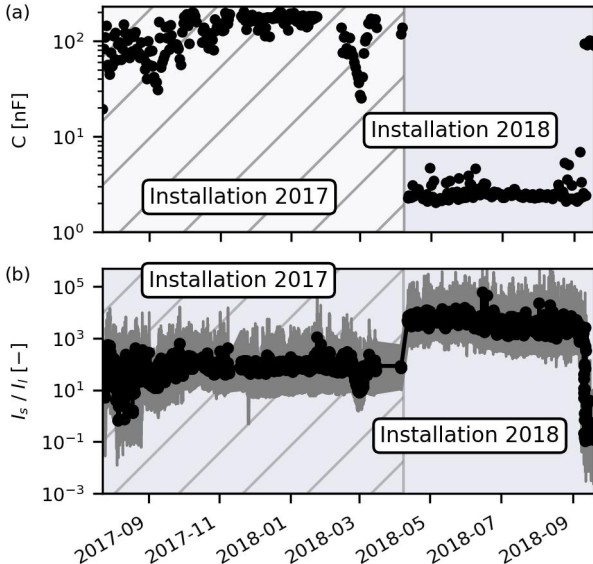

**Figure 10.** a) Evolution of daily mean cable capacitances at 1 kHz. b) Evolution of ratio between symmetric injection current and leakage current, $I_s/I_l$, at 1 kHz. Dark gray areas indicate min/max limits, and the hatched area indicates the time span in which measurements were conducted with the initial electrode and cable setup.

to extremely warm and dry weather. However, measurements in 2018 were terminated by equipment failure and as discussed below these highly variable contact resistances may also reflect unreliable measurements.

In addition to contact resistances, cable capacitances with respect to ground were also monitored on a daily basis. They showed a strong variation around a mean value of 100 nF in 2017 (Fig. 10a). Such large cable capacitances negatively affect

the accuracy of the sEIT measurements, as for example discussed in Zimmermann et al. (2019). In particular, high cable capacitances are associated with leakage currents that lead to an asymmetric current flow in the system, as well as additional current flow paths in the subsurface associated with the cable shielding. In principle, these effects can be accounted for in the FE modelling during the inversion, although this requires estimates of the distribution of the cable capacitances as well as extended forward modelling routines (e.g., Zimmermann et al., 2019). As with all correction and modeling procedures,

some uncertainty remains after correction. Therefore, it is desirable to reduce the cable capacitances and the associated leakage currents to the extent possible.

After the redesign in 2018, the capacitances were much lower and typically stable below 10 nF (Fig. 10a). We attribute this not only to the increased distance of the cables to the subsurface but also to the weather proofing using rain gutters, which prevented water from intruding into the cable bundles and thus from changing the dielectric properties between the cables.

The success of these measures can be seen in the ratio between symmetric excitation current $I_s$ and leakage current $I_l$, which increased and stabilised considerably after the redesign (Fig.10b). It is interesting to note that the time span near the end, associated with increased variability in contact resistances, also exhibits higher measured cable capacitances and very small





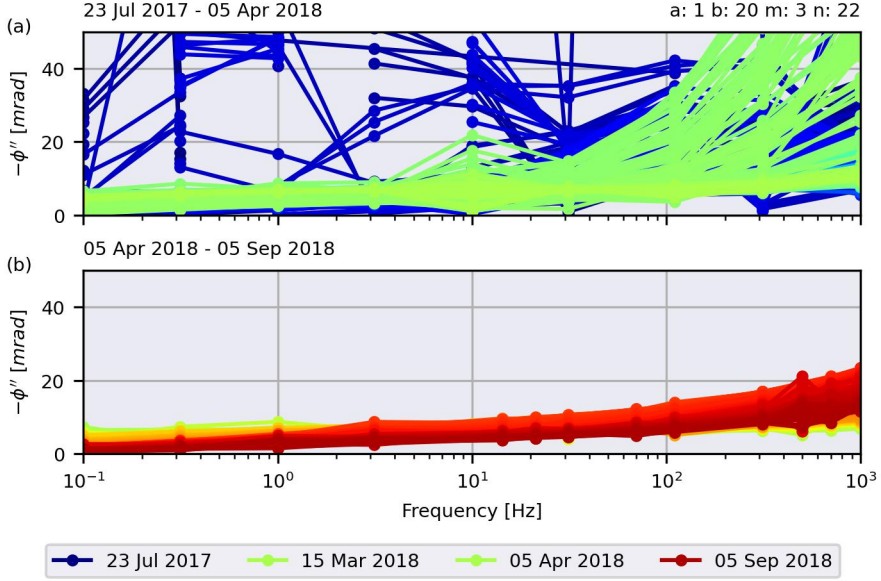

**Figure 11.** Temporal evolution of exemplary spectra divided into two time periods associated with different build stages of the measurement setup. The time of each measurement is colour-coded, with blue spectra indicating early measurement times, and red later ones. Measurements are not corrected for inductive effects.

ratios of symmetric to leakage currents. We interpret these very low ratios of the currents as the onset of the equipment failure that ultimately led to the termination of measurements. While low fractions of leakage currents could possibly be dealt with

using suitable modelling software, leakage currents in the same order of magnitude as the symmetric currents are a clear sign of unreliable equipment and measurements.

The observed improvements in the contact resistances and the cable capacitances correlated well with an improved behaviour of individual SIP spectra. Fig. 11 shows the temporal evolution of one selected measurement configuration divided into two time periods coinciding with the two build stages of the setup. Spurious data at the end of the monitoring period associated with

the failure of the equipment were excluded. The first time period exhibits quite erratic polarisation signatures with both usable and unusable SIP spectra, which is associated with variable contact resistances and leakage currents. The second time period starting after the redesign shows smooth and consistent SIP spectra. This highlights that low contact resistances and leakage currents are important for good data quality. However, it can also be seen that some useful measurements were obtained in the presence of higher contact resistances and leakage currents.

**5.4   sEIT imaging results**

The normalised phase RMS errors after the final inversion step are shown in Fig. 12 for 1 Hz and 1 kHz for the 2018 growing season and for different choices of the absolute phase measurement error. For both frequencies, a distinct change with time is





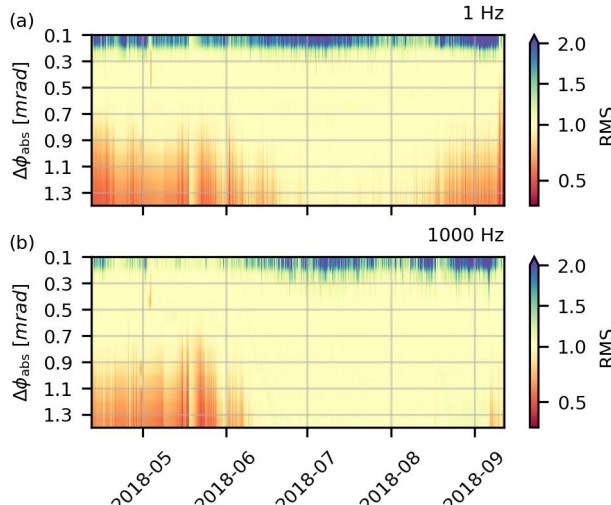

**Figure 12.** Normalised RMS error for the phase for the final iteration throughout the growing season 2018 for a) 1 Hz, and b) 1 kHz for different levels of absolute phase measurement error. Please note that the normalised RMS values for the magnitude are not shown.

observed, indicating a systematic temporal change in the noise characteristics of the data. However, there is an absolute phase error region between 0.3 and 0.9 mrad with RMS values near one for almost all times. The median values of all inversions with RMS values near one were used for further analysis. We also found that results for a phase error of 0.8 mrad showed very similar overall behaviour as the median results. Therefore, we occasionally also present results for this phase error in cases where extracting and computing the median results was too time consuming.

Exemplary inversion results for selected frequencies are presented in Fig. 13 for one day in May and one day in August 2018. We observed higher resistivity values in August compared to May, which is consistent with the increasing drying of the soil during summer. The resistivity magnitude for August 2018 (Fig. 13b) showed small-scale variability near the surface, sometimes consistent with high-phase anomalies. Analysis of these features showed that they are consistently present in certain time periods, and their position roughly corresponds to the position of the transparent plastic tubes embedded into the soil for optical root observations (for additional information on these rhizotubes, we refer to Cai et al., 2016). Additional possible explanations include selective drying and heterogeneity that cannot properly be modelled using the electrical forward model with the chosen mesh discretization. Also, a 2.5D modelling approach is employed which assumes a constant resistivity distribution perpendicular to the x-z plane shown in the tomograms. This assumption likely is occasionally violated in our experiments given the agricultural setting with heterogeneous plant growth and density, and the associated heterogeneous drying patterns.

The phase values generally show a smooth spatial distribution up to 1 kHz, but a distinct frequency dependence is apparent for higher frequencies (Fig. 13 c-j). The shallow anomalies discussed for resistivity results are also present, indicating that the underlying cause acts on both resistivity magnitude and phase. Spectral consistency of the imaging results becomes clearer in the intrinsic spectra recovered from the inversion results (Fig. (14)). Here, it needs to be emphasised again that the inversions

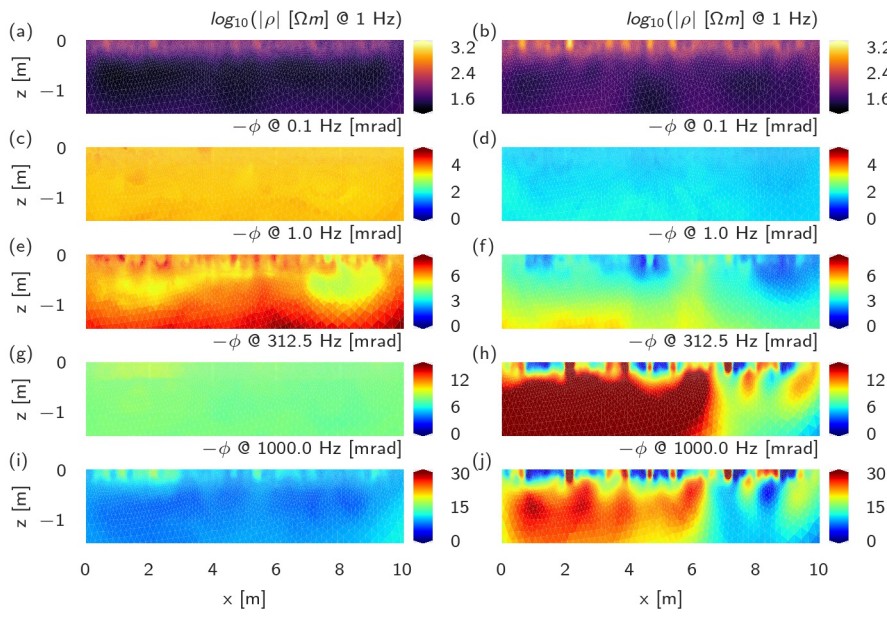

**Figure 13.** Temperature-corrected tomographic results for two time steps and selected frequencies. Left column: 1 May 2018, Right column: 1 August 2018. The inverted values are median values of all inversions in a 24 hour interval around the target time considering all inversion results with RMS values between 0.95 and 1.05. Note that the colour-bar limits vary for different frequencies.

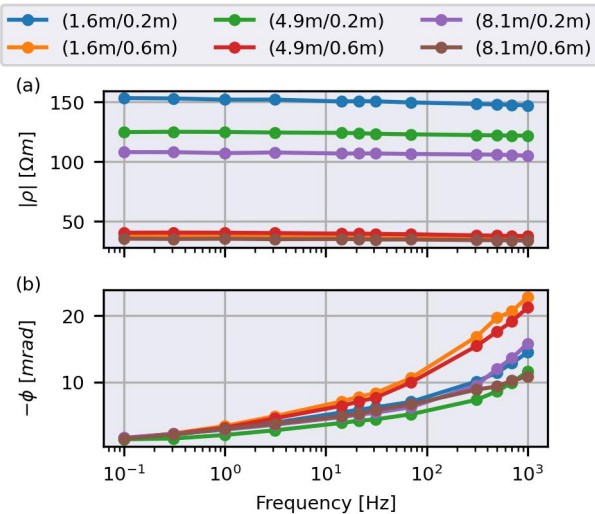

**Figure 14.** Intrinsic spectra extracted from tomographic analyses of all frequencies (August 2018). Labels denote the (x/z) position in the tomogram where the spectra were extracted. The spectra represent the median values of all inversions in a 24 hour interval around the target time considering all inversion results with RMS values between 0.95 and 1.05.

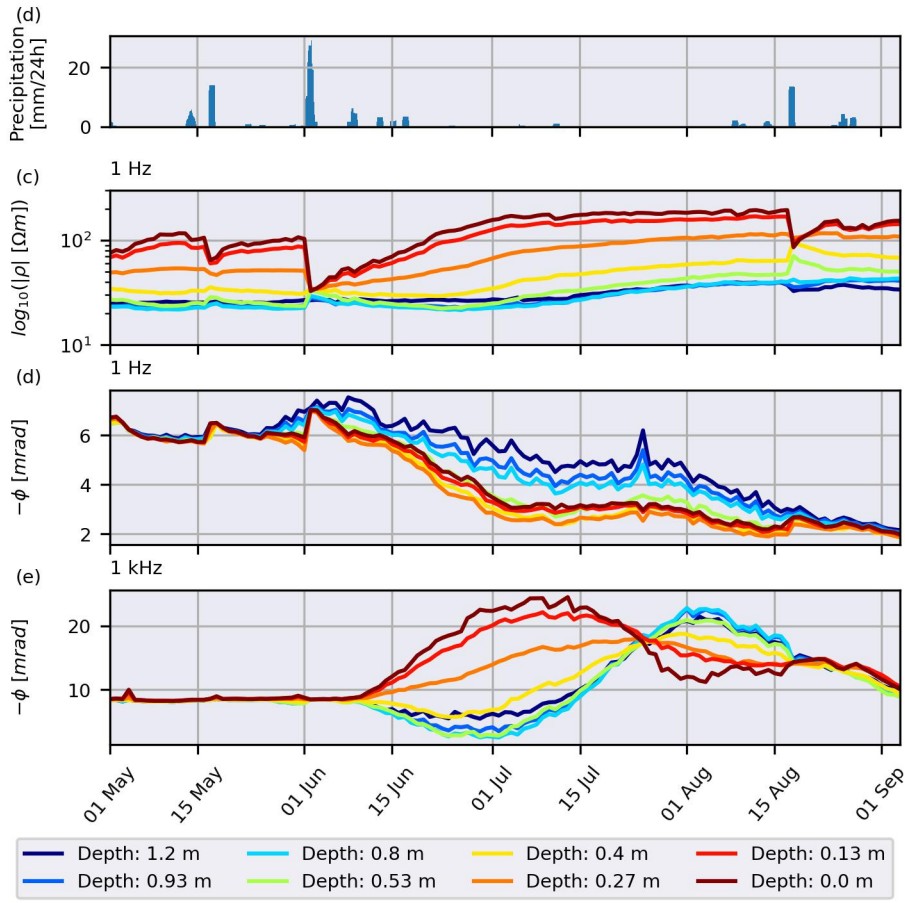

**Figure 15.** Temporal long-term development of the complex resistivity for the central region of the profile (3.25 - 6.5 m) for different depths. The electrical data points are median values of all pixels within a $\pm$ 5 cm range around the stated depth and all inversions in a 24 hour interval around the target time considering all inversion results with RMS values between 0.95 and 1.05. (a) Precipitation during the season 2018. Plotted is a rolling cumulative sum over 24 hour intervals. (b) Temporal development of resistivity magnitude at 1 Hz. (c) Temporal development of resistivity phase at 1 Hz. (d) Temporal development of resistivity phase at 1 kHz.

were conducted separately for each frequency. Thus, the smooth intrinsic spectra indicate a consistent convergence behaviour of the inversions for different frequencies for both resistivity magnitude and phase.

Finally, Fig. 15b-d presents the temporal development of the complex resistivity at different depths of the central region
of the tomograms ($x \in [3.25, 6.5]$ $m$). This region was selected because it is associated with the highest resolution for our electrode layout. The time series for both magnitude and phase at 1 Hz, and phase at 1 kHz, show consistent variation over time for different depths. It is interesting to note that the phase values at 1 Hz and at 1 kHz clearly show different temporal dynamics, hinting at a wide variety of information contained in these inversion results. As expected, large transient changes in the resistivity magnitude correlate well with precipitation events (compare Fig. 15a and 15b). The phase at 1 Hz also shows





a transient response associated with the precipitation-induced changes in the resistivity, but the phase at 1 kHz is much less affected by these events (compare Fig. 15c and 15d).

Phase values deeper in the soil (> 0.5 m depth) at 1 kHz showed a distinct minimum roughly at the beginning of July (Fig. 15d). While a detailed discussion of processes leading to this behaviour is beyond the scope of this study, we speculate that these low phase values are at least partially caused by inductive coupling effects that were not completely corrected. The

information for the larger depths of the tomograms stems from measurement configurations with larger geometric factors and thus smaller signal-to-noise ratios, which are more susceptible to inductive coupling. This hypothesis is supported by results obtained without correction for inductive coupling (Fig. A1d), where this minimum in phase is even more pronounced and even reaches physically implausible positive values.

Overall, it is found that three primary factors control the quality and consistency of the inversions results in this study:

the choice of error estimates, data filtering, and the inductive correction for high frequencies. While we are confident that our choices and procedures for each of these factors lead to robust and consistent results, we nonetheless note here that the qualitative behaviour of the results does not change in case of different choices. For example, the use of a relative phase error model, $\Delta\phi = \tilde{a}\,\phi + \tilde{b}$, provided similar inversion results (Fig. A2). The results also showed similar temporal and spatial behaviour if only measurement configurations are used that are present in 80 % of all time steps (Fig. A3).

## 6 Discussion

### 6.1 Technical aspects of the monitoring system

Our experience in setting up and maintaining an sEIT monitoring system highlights the complexity of this endeavour. While sEIT measurements in a laboratory setting are already challenging, applying this method to an even more complex and unsheltered field setting requires attention to many aspects to produce multi-dimensional electrical impedance data sets that are of

sufficient quality. Probably the most important aspect during operation was the monitoring of system health parameters such as contact resistances and cable capacitances with associated leakage currents. Monitoring and analysis of these parameters led us to redesign the electrode and cable installations, thereby improving the overall quality of measured data substantially. We anticipate that with time, when critical thresholds are better known for the system and specific field site, these health parameter checks could be directly integrated into the monitoring pipeline, and would allow for near-real time intervention in

case of decreasing data quality. Possible interventions could include fixing systematic or transient problems such as emerging short-circuits or water ingress into the system, as well as adapting shunt resistances and applied voltages to the measurement environment to improve data quality.

The results of this study also highlight that sEIT measurements cannot fully be treated in the quasi-static domain in which electromagnetic induction effects can simply be ignored. The extended correction procedure for mutual cable inductance effects

developed in this study significantly improves high-frequency data consistency, as shown in Fig. 8. We also found that data quality is acceptable up to 1 kHz based on raw data inspection (Fig. 11), inversion results (Fig. 13) and extracted intrinsic signatures (Fig. 14). Based on this finding, it seems prudent to evaluate inductive correction effects also for other frequency-





domain measurement devices, which would provide an alternative avenue of action for measurement scenarios where cable layouts cannot be optimised with respect to induction effects (e.g., Schmutz et al., 2014).

The monitoring and continuous measurement of cable capacitances to ground with associated leakage currents (Fig. 10) also provided important information that led us to redesign the set-up. This reduced the impact of capacitive coupling to a degree where it was not deemed necessary anymore to implement an explicit treatment of leakage capacitances and currents into the modeling and inversion framework, which is the alternative approach discussed in Zhao et al. (2013) and Zimmermann et al. (2019).

The measurement system was originally conceived as a laboratory system, and technical choices restrict the maximum excitation voltage against ground to $\pm 10$ V. While this aspect limits the injected current, and thus the attainable signal-to-noise ratio for field applications, it allows to operate the system without additional stringent security measures in field monitoring experiments. The low signal strength also directly leads to non-standard choices of measurement configurations, for which good signal-to-noise characteristics are favoured over optimal resolution. While the exact impact on attainable resolution is not

yet fully analysed, it has become clear that traditional configurations for polarisation measurements such as the dipole-dipole scheme lead to sub-optimal measurement results, as early experiments have shown (results not shown).

The large data sets generated by monitoring systems such as the one described in this study inevitably require a highly iterative work flow because processing often needs to be adapted when data are synthesised to address specific scientific questions. In retrospect, the use of a relational database helped tremendously in reducing the time it took to implement changes

in the processing structure and create new analysis steps, as the structure of the database allows for flexible data queries over large amounts of data. This flexibility is not present in traditional work flows in which data are often kept in separate files, and experience has shown that repeating processing steps without such a database is so time consuming that it becomes a bottleneck for scientific progress.

Geophysical measurements are rarely used in isolation, and thus other measurements must be scheduled to coincide with

them if various data sets are to be combined and compared. The use of the presented semi-automated measurement system simplifies the integration of sEIT measurements in existing research efforts, for example in agricultural research, and may ultimately lead to the integration of the system into decision making processes. Capturing temporally and spatially resolved monitoring data also allows the analysis of active processes in the subsurface, provided that they can be captured with suitable time resolution. With an estimated measurement interval of 2-3 hours, we can thus expect to resolve processes with periods of 4-

6 hours under optimal conditions. Yet, at this point no concerted effort was undertaken to optimise measurement operations for smaller measurement intervals. Given the ample literature on the optimisation of measurement configurations (e.g., Wilkinson et al., 2006), we see considerable room for improvement with regard to capturing faster processes.

## 6.2   Future developments

The sEIT monitoring system presented in this study is based on a laboratory design aimed at optimal measurement accuracy

and precision, but was not designed for ruggedness or large-scale applications. As also mentioned by Slater and Binley (2021), efforts have only recently started to design electrical measurement systems specifically for the task of long-term operation





with fixed setups. The adaption of the current sEIT measurements for robust long-term monitoring applications will require improving system resilience to harsh environments. This includes high humidity and large temperature dynamics. To a certain degree physical protection against vibrations and harsh handling could improve long-term usage and system installation and reliability. From an operational point of view, an integration with other environmental sensors is desirable. This would allow the system to react to external triggers such as extreme weather events or in response to temperature variations.

Another important point that will need to be addressed is the choice of electrode type. Steel electrodes were used here because they are relatively stable in time. However, even steel electrodes will chemically age due to corrosion and other chemical processes, and we also expect some degree of electrode polarisation at low frequencies when the electrode spacing is small. Therefore, future research should be directed into finding suitable electrodes for long-term sEIT measurements. While electrodes used for self-potential measurements could be evaluated (e.g., Petiau, 2000), they have a high interface impedance that hinders current injection, which is problematic for a system with low excitation voltages.

A major factor influencing tomographic inversion of electrical data sets is the choice of error parameters. While there have been investigations into the nature of data errors both for magnitude (e.g., LaBrecque et al., 1996; Koestel et al., 2008) and phase (e.g., Flores Orozco et al., 2012b), there is still considerable uncertainty in our knowledge of the specific error sources and how to deal with them. With respect to phase error estimates, two major phase error models have been used in the past, and it is still not clear which one should be favoured. Some studies have used absolute phase errors for their data (as we did in this study), implying that the phase errors do not strongly depend on overall signal strength (e.g., Kemna, 2000; Flores-Orozco et al., 2020). However, some studies using normal-reciprocal analysis have suggested a power-law dependency on resistance, $\Delta\phi = R^{-b}$ (e.g., Flores Orozco et al., 2012b). Future work should therefore also be directed into finding robust and, more importantly, consistent procedures to analyse electrical impedance monitoring data without relying on normal-reciprocal measurement for error quantification. Directly associated with this is the need to better quantify uncertainties of the resulting data sets. While first-order uncertainty estimates can be obtained through numerical simulations for multi-frequency impedance imaging (e.g., Weigand et al., 2017), quantification of the spatial parameter uncertainty would probably require the application of stochastic inversion frameworks, which have recently been established for DC geoelectrics (e.g., Galetti and Curtis, 2018).

The inversion of multi-dimensional sEIT data involves a large parameter space that makes it hard to obtain robust and consistent results. Therefore, future work should aim to further improve the temporal and spectral consistency of the inversion results, for example by applying smoothness constraints along the spectral dimension (e.g., Kemna et al., 2014; Günther and Martin, 2016). Alternatively, the careful integration of sEIT data in structural or process-based models could decrease the size of the parameter space to improve the consistency of the inversion results.

## 7 Conclusions

In this study, we presented a semi-automatic sEIT monitoring system aimed at the small field scale. Multiple data quality parameters such as contact resistances, cable capacitances, and resulting leakage currents were routinely monitored along





with the actual sEIT measurements. By analysing these quality parameters, significant improvements in system reliability and imaging quality could be achieved, for example by adapting measurement configurations, decreasing cable-to-soil coupling, and by improving data filtering before tomographic analysis. Additionally, inductive coupling effects between cables were corrected using a novel procedure that combines numerical modelling and calibration measurements, leading to reliable data up to 1 kHz. We conclude that data quality control should be an integral part of sEIT monitoring activities and special attention

should be given to capacitive and inductive coupling effects and their potential remediation. With suitable knowledge of the internal workings of other sEIT systems, methods for correcting inductive coupling effects should be applicable also to other systems. The achieved data quality, consistency, and spectral bandwidth over the time span of multiple months, with multiple measurements per day, signifies an important step in establishing the sEIT method as a suitable monitoring tool not only for near-surface geophysical but also for biogeophysical applications.

*Code and data availability.* Scripts and data to reproduce all images presented in this study are available on reasonable request from the corresponding author.

## Appendix A: Temporal Development of Inversion Results

In this appendix, the temporal development of resistivity magnitude and phase values at 1 and 1 kHz extracted from the imaging results is presented for alternative data processing strategies. Results for data without any inductive correction applied
are shown in Fig. A1. Inversions conducted using the same data as presented in the main text, but a different phase error model with a relative phase error model, $\Delta\phi = \tilde{a}\phi + \tilde{b}$, are shown in Fig. A2 for $\tilde{a} = 0.05$ and $\tilde{b} = 0.3$ mrad. Finally, inversions where data were filtered using a temporal filter that only retained measurement configurations present in at least 80 % of all time steps are shown in Fig. A3.

While small temporal details vary between the phase responses of all three scenarios we note that the overall patterns stay
the same, even for the case where data was additionally filtered to only include data points present in at least 80% of the time steps ( Fig. A3). The only significant variation in the results can be observed for phase values at 1kHz, highlighting the effect of a proper inductive correction procedure (compare Figs. 15 and A1). We therefore conclude that while a proper treatment of data filtering procedures and inversion settings is crucial for quantitative approaches, qualitative results should be quite robust for a wide range of different processing choices.

*Author contributions.* AK, EZ, SH initiated planning and funding of the research. AK, EZ, SH, MW planned the technical realization of the monitoring system. EZ, MW installed and operated the system. EZ, MW, VM planned and conducted the inductive calibration measurements. All authors discussed the results and contributed to the manuscript.

*Competing interests.* The authors declare no competing interests.

*Acknowledgements.* This work was funded by the German Research Foundation (Deutsche Forschungsgemeinschaft, DFG) under Germany's Excellence Strategy, EXC-2070 - 390732324 – PhenoRob, and the SFB/ TR32 "Patterns in Soil-Vegetation-Atmosphere Systems: monitoring, modelling, and data assimilation". The rhizotron facility is supported by TERENO (Terrestrial Environmental Observations) funded by the Helmholtz-Gemeinschaft. We are grateful to Walter Glaas for ongoing technical support for the sEIT monitoring system, and Andrea Schnepf, Anja Klotzsche, Normen Hermes and Dino Schmitz for their support with the maintenance of the rhizotron facility. We also thank
Andreas Dreist for technical support and maintenance of the processing and archiving facilities at the University of Bonn.

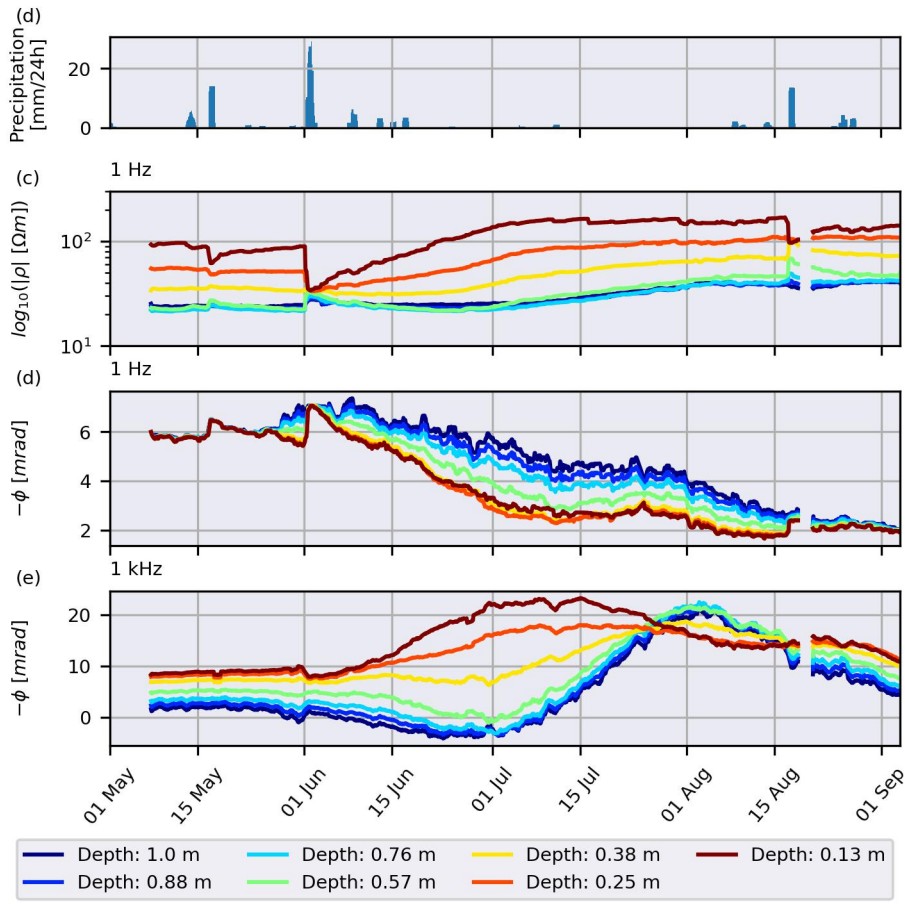

**Figure A1.** Temporal long-term evolution of tomographic imaging results. No inductive correction was applied to the measured impedance spectra. Shown are rolling median values for the central plot of the profile (3.25 - 6.5 m) over 24 hour windows for different depths. An absolute phase error of 0.8 mrad was assumed for all inversions. (a) Precipitation during the season 2018. Plotted is a rolling cumulative sum over 24 hour intervals. (b) Time evolution of resistivity magnitude at 1 Hz. (c) Time evolution of resistivity phase at 1 Hz. (d) Time evolution of resistivity phase at 1 kHz.



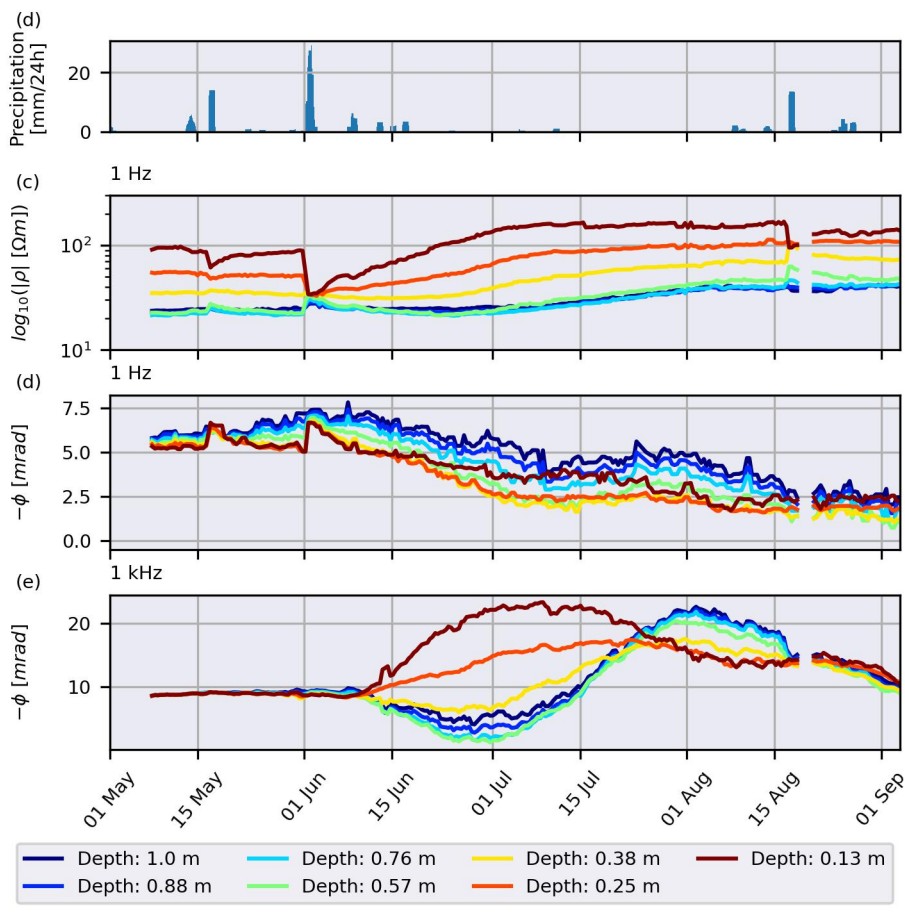

**Figure A2.** Temporal long-term evolution of central part (3.25 - 6.5 m) of the profile. Phase inversions were carried out with a relative phase error of 5 %, with a stabilizing absolute phase error of 0.3 mrad. Shown are rolling median values with a window width of 24 hours. (a) Precipitation during the season 2018. Plotted is a rolling cumulative sum over 24 hour intervals. (b) Evolution of resistivity magnitude values at 1 Hz for different depths. (c) Evolution of phase values at 1 Hz for different depths. (d) Evolution of phase values at 1 kHz for different depths.



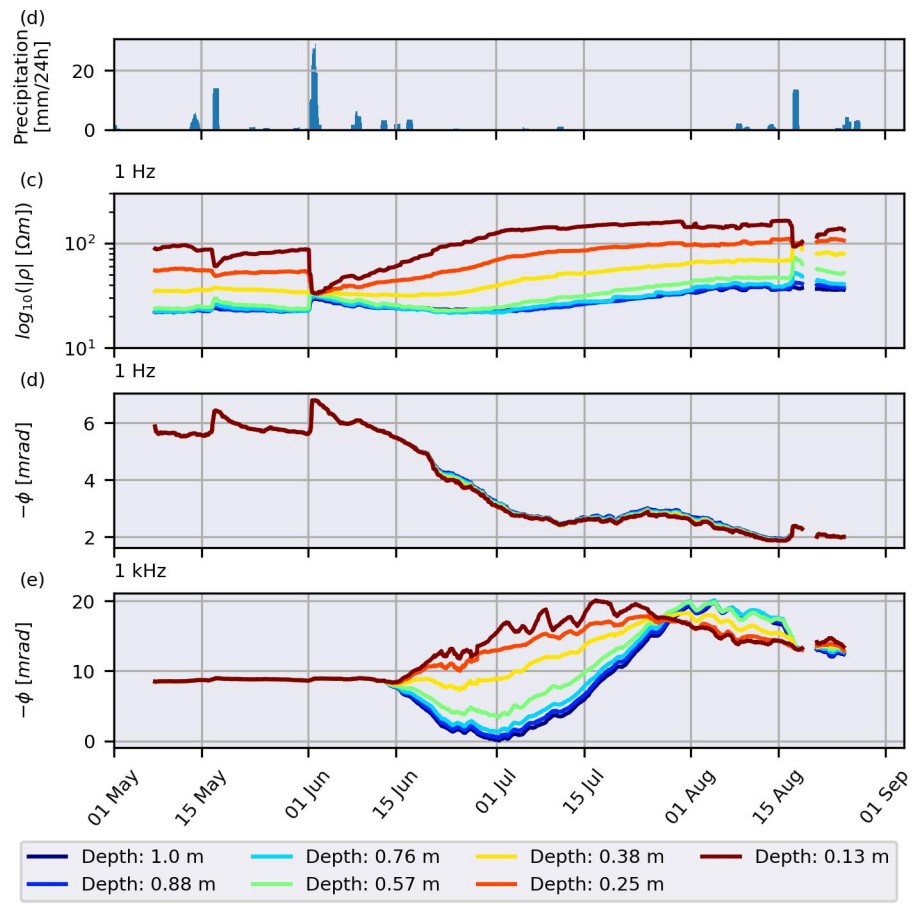

**Figure A3.** Temporal long-term evolution of tomographic imaging results for the time-filtered data set which uses only measurement configurations present in at least 80 % of the time steps. Shown is data from the central plot of the profile (3.25 - 6.5 m) for different depths. An absolute phase error of 0.8 mrad was assumed for all inversions. Shown are rolling median values with a window width of 24 hours. (a) Precipitation during the season 2018. Plotted is a rolling cumulative sum over 24 hour intervals. (b) Time evolution of resistivity magnitude at 1 Hz. (c) Time evolution of resistivity phase at 1 Hz. (d) Time evolution of resistivity phase at 1 kHz.





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
