# Peer review of "Design and operation of a long-term monitoring system for spectral electrical impedance tomography (sEIT)"

_Geoscientific Instrumentation, Methods and Data Systems, 2021_

## Author Comment (AC1)

**Responses to reviews of manuscript gi-2021-36**

Please find below our detailed response to the reviewer comments as well as additional improvements that have been made during the revision.

**1 General changes**

- Fig. 14 was regenerated from the newest inversion results presented in the rest of the manuscript. During revision it was found that the presented spectra were generated from a previous inversion run with slightly degraded quality of the inductive correction procedures.

[Figure]

Figure 1: Old Fig. 14                    Figure 2: New Fig. 14

**2 Reponses to reviewer # 1 (Andrew Binley)**

> *Comment 1*:  *I read this manuscript with great interest. Although field-based monitoring of DC resistivity has been around for several years, recent instrument developments have resulted in this being now widely available.  Several manufacturers now offer the capability for relatively long-term monitoring of DC resistivity. However, the monitoring of induced polarization (IP) presents new challenges, and when one considers spectral IP (SIP) then many additional problems emerge. Accurate measurement of SIP in a controlled laboratory setting is not trivial. To translate this to the field and consider measurements over a period of time is immensely challenging. Several researchers have argued that there could be useful information captured in a measure of electrical polarization that can help improve our understanding of subsurface processes and properties. We still do not fully understand many of the controls of electrical polarization in geo materials but as we improve knowledge of such controls there will be, no doubt, a demand for SIP monitoring solutions.  It is, therefore, good to see that the authors have taken steps to develop such a system and document their initial findings. An SIP measurement is easy to make.  An accurate SIP measurement is not. Without a clear understanding of some of the issues that researchers need to consider (as outlined in this manuscript) there is a danger that unreliable measurements are obtained, and inaccurate interpretations made.*
> *The manuscript essentially documents the authors' attempts to implement a new system for temporal monitoring of SIP at a study site in Germany – the Selhausen rhizotron facility. The authors explain clearly their rationale for their design, explain how they conducted the measurements over a period of several months and the steps taken to assess data quality and filter data.  They also show resultant images (and aspects of images) that result from the time-monitored data. The authors admit that their system is far from perfect but offer some suggestions for improvement.*

Thank you for these encouraging comments.  We fully agree that SIP and sEIT field applications, especially monitoring attempts, are still at the very edge of current methodological and procedural understanding.  We also agree that quite a few problems need to be solved until the method can be easily applied. This manuscript therefore not only presents the specific monitoring system, but also attempts to highlight some of the many challenges that we need to deal with.

> *Comment 2*:  *Overall, the manuscript is clearly written and includes some carefully prepared illustrations. A few of the figures would benefit from revision (see specific comments later).*

Thanks, we tried to address all your comments below.

> **Comment 4**: *The title refers to "long term monitoring". This is, in my view, an over-used and often inappropriate phrase. I'm not sure that monitoring over several months is really "long term". I am being pedantic here but if the authors are to use such a term then they should define what they mean by it.*

The abstract now clarifies that we present a system designed for multi-month or multi-year monitoring periods. We also clarify in the introduction that the system is capable of not only multi-month, but also multi-year operations.

> **Comment 5**: *My main concern about the work detailed in the manuscript is that the field experiment is not well controlled. What I mean by that is that the authors collected data and produced images but we have no idea what processes occurred in the field setting. There is mention that SIP can tell us something about soil water and even plant roots but we are not given any information about the soils under study, nothing about the internal states and nothing about what plants are grown (if any) and their characteristics. It is not a controlled experiment and I believe that this was a major oversight since we are unable to assess whether the signals in the resultant images make sense. The authors show some time-lapse results but only report on precipitation as a complementary observation. If this is a "facility" then surely other states were observed. Why are they not reported? Even time-series of soil moisture at several depths would help. What is the difference between plots 1, 2 and 3 in the facility? I realise that the authors wish to focus on instrumentation aspects and not be distracted by internal processes in the soil but without any independent observations many comments on influences are speculative. A much more appropriate setup (in my view) would have been to have ensure much greater control of the boundary conditions (infiltration, evaporation) and internal states (moisture, temperature, presence/absence of plant roots, etc.). Nevertheless, the findings are very useful. I just think that the authors need to provide more information to allow the reader to get a better understanding of properties and dynamics of states that are controlling the observed electrical responses.*

The reviewer raises a valid point when the study is viewed as a scientific investigation of analysing root systems with SIP/sEIT. However, as clearly stated in the abstract and introduction (and also recognised by the reviewer in his comment), this manuscript aims to discuss the technical aspects of an sEIT monitoring system. Adding more environmental information would increase the scope of the manuscript considerably, while compromising on structure and completeness. The experiment under investigation indeed has much more soil and plant information available, but that information was deliberately left out due to size considerations and with the aim to focus the discussion on the technical details. Results are presented to show spatial, spectral, and temporal consistency in the results, not attempt a biogeophysical subsurface characterisation, which would in turn require much more background and further analysis. It should be noted here that the further analysis and

interpretation of the inversion results will be the subject of a second, dedicated manuscript.

We recognise that reader expectations are important and thus added corresponding notes to the results and discussion sections, making clear that the technical aspects of the system are the scope of this text, also corresponding to the aims of the GI journal.

> **Comment 6**: *The authors explain in their discussion that the system was originally developed for laboratory scale investigations. This helps explain why they use low (+/- 9V) voltages for the source. I think it would help if there is a bit more discussion on what challenges (if any) in moving to a higher voltage source since this would help transfer this approach to a broader range of studies/applications.*

A corresponding paragraph was added to the discussion (Sect. 6.2), explaining why the existing hardware cannot be extended to excitation voltages above $\pm$ 10 Volts.

> **Comment 7**: *I understand the logic for selecting long dipole lengths to cope with the relatively low signal source. However, it seems that the shortest potential dipole is 3.5m, which seems very long to me given that you infer something about processes at very shallow depths. The images in Figure 13 are to depths of 1.2m. What exactly is your resolution capability? I think that at the very least you need to show some sensitivity maps. I also think that you need to be careful in discussing the inferred high resolution variation in (complex) resistivity given the electrode configuration and also discuss this further.*

Indeed some more discussion is helpful here, thank you for pointing this out.

- The measurement configurations were chosen from a complete configuration set, which theoretically contains all information on the subsurface that can be gained for a given number of electrodes and electrode spacing (e.g., Xu and Noel, 1993). A complete data set allows the generation of arbitrary four-point measurements by means of superposition. Yet, by generating arbitrary four-point measurements by means of superposition also implies that the resulting data error increases by means of error propagation. This often restricts real-world applications of the principle due to large resulting data errors.

  Therefore, given enough large dipoles also small structures can be resolved (noise issues are discussed below). As an example we conducted a simple synthetic study where we compared a full skip-0 dipole-dipole measurement configuration with one configuration similar to the one used in this study, consisting only of large dipoles with electrode separations of 19 or 21 electrodes. The results clearly show that the large dipole configurations can produce similar, if not better, results, provided a suitable noise environment:

[Figure]

Figure 3: Synthetic inversion study showing the capability of large-dipole configurations to resolve small features. (a) Forward model used to generate synthetic measurement data (b) Inversion result based on synthetic data generated for a full skip-0 dipole-dipole measurement scheme. (c) Inversion result based on synthetic data generated for a complete data set consisting of dipoles with lengths of 19 and 21 electrode separations.

Regarding noise pollution and error propagation, we note that on one hand we added additional four-point configurations to the analysis to alleviate influences of noise propagation. Also, we hypothesise that linear error propagation does not increase resulting data errors faster than geometric factors increase (associated with a corresponding decrease in signal-to-noise ratio).

- Finally we note that this issue should still be investigated in more detail, as is the case with quite a lot of aspects of todays geoelectrical tomographic measurements. These investigations are left to future studies.

The discussion section 6.2 was modified to discuss future work on measurement configurations.

- We modified section 4.4 (Measurement Configurations) accordingly and explain that we actually recover a complete data set with additional redundant configurations.

> **Comment 8**: *Monitoring of SIP data will inevitably lead to periods of data collection with different acceptable measurement configurations (as illustrated in Figure 6). I think that this aspect needs some discussion. An image produced from 1500 measurements surely has different resolution capability than one based on 1000 measurements. While this is an issue for other monitoring techniques, SIP is particularly prone to dataset size dynamics.*

- Section 4.4 (Measurement configurations) was modified and we now explain in more detail that the employed measurement configurations are not only complete in the sense of information content (Xu and Noel, 1993), but also that redundant configurations were used to minimize the effect of data filtering.

- We added coverage information for space and time in appendix A of the manuscript. The coverage data implies that the captured information does not show strong variations over time.

- While the coverage information suggests that data filtering does not significantly change the resolving capabilities of our measurements, we acknowledge the limits of solely looking at the coverage (sensitivity) and refer to the future for a more in-depth analysis in the discussion.

> **Comment 9**: *Line 4. It would be useful to state the frequency range in the abstract since impedance spectroscopy means different things to different readers.*

Done

> **Comment 10**: *Line 4. Change "polarization" to "polarisation" to be consistent throughout.*

Done

> **Comment 11**: *Line 9. Define "long-term".*

Done

> **Comment 12**:  *Line 14. What "core"? I can work out what the authors mean but some readers may be confused by this.*

We replaced 'core measurement system' with 'data acquisition system'.

> **Comment 13**:  *Line 26. They may not always be "independent".*

The paragraph was modified to point out ambiguities in the data interpretation.

> **Comment 14**:  *Line 73. The Telford et al. (1990) reference is inappropriate regarding a statement on EIT. It would be unethical of me to propose citation of my own work but Binley and Slater (2020) [Resistivity and Induced Polarization. Theory and Applications to the Near-Surface Earth, Cambridge University Press] is a much more appropriate reference.*

Corrected.

> **Comment 15**:  *Line 87. Explain "small levels of systematic variations".*

Both normal and reciprocal measurements must contain only low systematic (non-Gaussian) noise components in order for the normal-reciprocal analysis to yield good (non-biased) results. We modified the text to state this explicitly and added a note that different signal-to-noise ratios occur for differently-sized current and voltage dipoles.

> **Comment 16**:  *Line 88. Explain "heuristics must be employed".*

Without sound means to determine the actual error parameters (such as normal-reciprocal analysis) we must resort to other approaches to determine error parameters for the inversion. The text was modified and now the corresponding sentence reads: "In such cases, other, often empirical, approaches must be employed to find suitable data error estimates for the inversion. "

> **Comment 17**:  *Line 94. Don't mix precision and accuracy.*

The text was modified and we now only point out that L1-norm inversions usually over-smooth images and are not useful for quantitative data analysis.

> **Comment 18**:  *Line 95. Explain "filtering are the only options" – you are implying that your (our) normal processing of errors is not useful in determining quantitative estimates of values, and yet you (we) have used this in this way many times.*

We here are referring to systematic noise components that are non-Gaussian in nature. This type of error bias the imaging results and cannot be described by error models such as those based on normal-reciprocal analysis.

The text was modified accordingly.

> **Comment 19**:  *Line 117. Make it clear that this is a field site, not a lab setup.*

Done

> **Comment 20**:  *Line 118. "basement" is an odd and ambiguous term. Some explanation is needed. Perhaps a photo of the setup would help.*

The term was replaced by "wooden container in excavated pit", consistent with Cai et al. (2016).

> **Comment 20**:  *Figure 1. How far away is the electrode array from the edge? It looks to be about 5m. This needs to be covered so that we can understand any 3D effects.*

Fig. 1 was modified to show distances to the borders of the field (2.4 and 4.6 m).

[Figure]

Figure 4: Old Fig. 1          Figure 5: New Fig. 1

**Comment 21**:  *Line 129. Remove "actual". It is redundant.*

Done

**Comment 22**:  *Line 144. What "cable effects"?*

- Placing the amplifier near the electrode reduces capacitive load on the system (Zimmermann et al., 2008).

- The text was modified accordingly.

**Comment 23**:  *Line 147. Comment on propagation of errors with superposition.*

 Done. Sect. 4.1 was modified accordingly. We now point out that the error increases due to the two-component error propagation, but is drastically reduced by optimising the common-mode error in the individual three-point measurements (see also Zimmermann et al. (2008)).

**Comment 24**:   *Line 161.  The shunt resistor has been selected for this particular study.  How would this be selected for others?*

For our system a shunt resistance matching the series resistance of electrode-soil contact resistance and soil resistance will lead to an approximately equal amplitudes of current and voltage signals.

The text was slightly modified accordingly.

> **Comment 25**: *Line 172. It might be useful to reference the published work that explains the rationale for accepting the point electrode assumption here.*

We added a reference to Rücker and Günther (2011).

> **Comment 26**: *Line 176. So the electrode is at 10-15cm depth. Is this modelled as a buried electrode? If it is not (and assumed that the electrode is a point at the surface) then what are the implications?*

Electrodes were modeled in 12.5 cm depth, corresponding to a position of the point electrode at 50 % length of the electrode. Given the uncertainties associated with field surface microtopography we feel that this is close enough to the 60% suggested by Rücker and Günther (2011).

The text was modified to include the modeled electrode depth.

> **Comment 27**: *Line 312. Typo in figure number.*

fixed

> **Comment 28**: *Line 370. This is not specific to Hayley et al.(2007) – it is the standard approach that many adopt. All you are doing is correcting the cell values as anyone would do. Hayley actually proposed a more sophisticated method that corrected data not inverted values.*

- Thank you for pointing out the generality of the approach. Yet, equation 4 in Hayley is exactly our equation 1. In addition, Hayley et al. (2007) corrected conductivity tomograms exactly as we did.

- We clarified the text that the approach is quite general and added citations to Sen and Goode (1992); Hayashi (2004).

**Comment 29**:  *Line 377. Over what depths to you have soil temperature measurements?*

Temperature measurements were installed at depths of 10, 20, 40, 60, 80, 120 cm at 9 horizontal positions. The text in Sect. 4.8 was updated to include the depths.

**Comment 30**:  *Line 429. They increased but also decreased – what is the cause of this? You cannot justify this as due to soil drying. It looks more like an instrumentation fault to me.*

Indeed we mentioned faulty instrumentation as the second possible cause of the observed changes. The text was modified to be more specific ('increases not in values, but in variability and maximum values'). Also, random resistance fluctuations at the soil-electrode interface are mentioned as a possible cause for the observed dynamics, but instrumentation failure is mentioned as the most probable cause.

**Comment 31**:  *Figure 11. I can't follow what is being plotted here - there are two green dates. Some figure reworking needed and a clearer caption.*

- The figure caption was reworked.

- The figure was slightly reworked:

[Figure]

Figure 6: Old Fig. 11                              Figure 7: New Fig. 11

> **Comment 32**: *Figure 12. Delta phi is not defined. I assume it is the phase error. So, for a given point in time do we have a range of delta phi (and RMS) for the range of frequencies? Is this what is being shown here? More explanation needed.*

- $\Delta\phi$ is now defined in the text and figure caption.
- We added an introductory sentence explaining that a parameter search was conducted for each data set and also reference back to section 4.7 (Data inversion), where the parameter search for an absolute data error is explained in detail.

> **Comment 33**: *Line 469. This is an example of where information on plants being grown is needed. We cannot compare May and August events without knowing something about the state of the plants.*

- We added a paragraph at the beginning of the result section to clarify that the results are meant to merely show consistency of the data, not provide a full and detailed analysis and interpretation.

> **Comment 34**: *Line 476. You don't give any information on variability in y.*

As with most applications of the 2.5D inversion approach we have no information on subsurface variability in the y direction. While agricultural settings always come with uncertainty with regard to soil and plant structure, our study was conducted on a controlled field site (e.g., Cai et al., 2016), which suggests that y-variability is relatively low.

We modified the corresponding text in this regard, also noting that we have at least 2.4 meters distance to each border on the y-axis.

> **Comment 35**: *Line 481. I don't think that "intrinsic" is the right term here.*

While we agree that other terms could possibly be more specific, we decided to keep the term for consistency with previous work (Weigand and Kemna, 2017; Weigand et al., 2017; Weigand and Kemna, 2019).

> **Comment 36**: *Line 481. Are the selected spectra typical? Or very carefully selected?*

While we did not check all locations and time steps, the majority of investigated spectra were smooth similar to those shown in Fig. 14. The shape of the spectra changed over time.

Appendix B was added to the manuscript, showing SIP spectra for the months May, June, July, August and September.

> **Comment 37**: *Figure 15. We need to see much more than rainfall!*

If an in-depth structural and process-orientated analysis of the results would be attempted, then we would agree. However, this manuscript is focussed on the technical aspects of the monitoring system and Fig. 15 primarily serves to show the consistency of the data, manifesting in consistent temporal evolution of independently analysed tomographic results.

We added a sentence to the discussion of Fig. 15 that reiterates that a detailed analysis is beyond the scope of this study.

> **Comment 38**: *Line 485. Are you showing an average? If so, why? It defeats the purpose of imaging.*

- This very much depends on the target to investigate: If a decimeter-scale investigation of the subsurface is desired, then yes, averaging would defeat the purpose. However, we still retain any vertical resolution of the setup, and thus lateral averaging to the size of the plots (each with a different treatment) is a compromise between lateral resolution and improving signal-to-noise ratio.

- In short: It depends on what we want to look at. Lateral averaging here helps to highlight long-term (multi-month, multi-year) stability of the data.

> **Comment 39**: *Line 488. I agree but what this paper lacks is a clear statement of why this might be useful.*

Referring to our answer to comment 37, we think this is not in the scope of the study beyond the fact that we highlight that we get different information at different frequencies.

The text was slightly modified to better reflect that 1 Hz and 1 kHz probably contain complementary information.

> **Comment 40**:  *Line 495. This suggests that you need to plot some image appraisal information. You state just a bit earlier that the phase data gives information and now you say that some of this is unreliable.*

- We added coverage information for space and time in appendix A of the manuscript. Coverage data imply that the captured information does not show strong variations over time.

- The nature of the measurements is that they have some uncertainty attached to them. As such, containing information and being uncertain is not contradictory.

> **Comment 41**:  **SUMMARY** *In summary, I think that this is a valuable piece of work that illustrates some of the challenges in monitoring SIP, and provides very useful insight into steps that are needed to assess and improve data quality. I sense that this will prove to me a useful reference article. My major concern is that the authors do not know what processes are operating in the soil (or at least they don't show any evidence that they do) and so we do not know what is right and what is wrong. I also have some concerns about the inferred resolution of the resulting images. These issues can be addressed with relatively minor revision.*
>
> *Andrew Binley*

**3 Responses to Reviewer # 2**

*Introduction*:   *This is a very interesting paper describing the test experiment for the small-scale sEIT monitoring system, in which several data quality parameters such as contact resistances, cable capacitances, and resulting leakage currents were monitored along with the actual sEIT measurements. The authors demonstrated how detailed analysing of these quality parameters was used to improve significantly the measurement system setup in real monitoring experiment.  They developed the ways to improve the system's reliability and also imaging quality.  One of the advantages of the paper is investigating inductive coupling effects between cables and development of a novel technical solution and procedure for correction of these effects that is combined with the calibration procedure.  The authors managed to demonstrate the performanse of the system under real field conditions in a small-scale biogeophysical experiment. Their foundings and recommendations concerning materials for electrodes and other technical details of their equipment are very valuable.*

*Comment 1*:   *However, I think that the paper still requires some improvement. Here are some of my comments: 1) In Chapter 2, lines 90-97 you wrote that L1-norm inversion is a well-known mean to perform robust inversion of the data with large amount of noise or/and outliers. It seems that your data represent exactly this case.*

While the L1-norm (applied to the data misfit) inversion is indeed more robust with regard to outliers, it still requires knowledge of the underlying distribution of the data errors.  In the case the L1-norm inversion assumes a exp-random error distribution (Menke, 2012), although other robust norms have also been used, implying other random distributions of the data.  While this distribution has longer tails, and thus can accommodate larger outliers, a quantitative and non-biased inversion still requires the data errors to originate from the exp-distribution. This is not the case here. One additional aspect is that when the L1-norm inversion is used to measure the data misfit, experience has shown that the image resolution may be significantly reduced when data outliers are few, leading to overly smooth images that are typically only loosely dependent on the choice of data error estimates.

> **Comment2**: *It is not true that such a scheme is not providing the way to evaluate uncertainty of the results, as one can always use general Bayesian formulation of inverse problem and obtain the uncertainty estimate in the form of a-posteriori probability density function (see Tarantola, 1987). For me it is not clear why you still decided to follow the general inversion scheme described in equation (1), because it seems that it is not suitable for your type of data, as the regularization implemented in (1) was clearly not sufficient to stabilize your solution. So what was the reason that you still followed the procedure that is well suited for Gaussian distribution of error in the data only?*

Bayesian approaches also require prior information on the shape of the error distribution. As we do not have detail information about the used error distributions, we usually assume a normally-distributed error source and try to remove all outliers that would skew the error distributions too much. This rough estimation of the underlying error distribution is inherent to both the least-squares-based inversion used in our study, and Bayesian approaches.

Regarding the regularization we note that the inversion is successfully stabilised by the smoothness.

Our efforts in this study were geared toward generating robust and consistent data and images **over time**. The rationale here is: If we properly remove outliers and come up with appropriate data error estimates, then inversion results should show temporal consistency, even when each time-step is inverted independently.

> **Comment 3**: *2) As a result, you had to use a large amount of empirically defined parameters for additional filtering the data, as it is described in Chapter 4.6. I think that justification for using them is quite week and should be described more detail. For example, what happens if a threshold in (8) is 3.05, but not 3, or can the threshold in (9) be 9.56, for example?*

We agree that the large number of empirically derived parameters is suboptimal, and future work should go into reducing the number of those parameters required. This is now mentioned in section 4.6 (Data Processing)

However, we note here that the actual choice of the used filter parameters does not significantly change the overall results. All main features of the images remain even if a greatly reduced data set is used (see Fig. C3, which shows results where only measurement configurations were used that were present in 80 % of the time steps).

Finally, we also would like to point out that the use of empirical parameters is often an inherent problem of geoelectrical analysis, but rarely discussed in the literature. One example would be the choice of measurement configuration and associated filtering: By choosing to only analyse a given set of measurement configurations, for instance with relatively large signal-to-noise ratios, we would reduce the need to apply any data filtering processes. However, the actual empirical decision is still made when the measurement configurations are decided upon.

In our case the system measures potentials towards system ground, allowing us to compute a large number of measurement configurations, irrespective of their actual usefulness. As such we move the empirical decision making from deciding which configurations to use (we just generated a huge amount of configurations) to filtering (empirical filter parameters).

> **Comment 4**: *3) Looking at Figure 6, one see a clear temporal variations in the amount of data points left after filtering. In particular, application of SMOOTHENESS filters removed half of the data after 24.06. Do you have any explanations for this?*

- At this point we cannot give any definitive answer to your question. However, the SMOOTHENESS filter is basically a measure of how much a given spectrum deviates from a horizontal line. Looking at Fig. 11b of our manuscript you can see that the spectra change their shape over time and their slope increases (compare early, green, curves with late, red, curves). As such, for a fixed threshold we can assume that slightly smoother spectra will be filtered by the SMOOTHENESS filter at later times due to the larger slope and thus associated larger baseline roughness.

- As a first-order image appraisal tool we included a short analysis of the spatial and temporal distribution of the cumulated sensitivity in appendix 1.

- Section 4.6 was modified to now include a statement that, despite the variations in filtered data points, coverage remained roughly the same over the measurement time (see Fig. A1).

**References**

Blanchy, G., Virlet, N., Sadeghi-Tehran, P., Watts, C., Hawkesford, M., Whalley, W., and Binley, A.: Time-intensive geoelectrical monitoring under winter wheat, Near Surface Geophysics, 18, 413–425, doi: 10.1002/nsg.12107, 2020.

Cai, G., Vanderborght, J., Klotzsche, A., van der Kruk, J., Neumann, J., Hermes, N., and Vereecken, H.: Construction of minirhizotron facilities for investigating root zone processes, Vadose Zone Journal, 15, doi: 10.2136/vzj2016.05.0043, 2016.

Hayashi, M.: Temperature-electrical conductivity relation of water for environmental monitoring and geophysical data inversion, Environmental monitoring and assessment, 96, 119–128, doi: B:EMAS.0000031719.83065.68, 2004.

Hayley, K., Bentley, L., Gharibi, M., and Nightingale, M.: Low temperature dependence of electrical resistivity: Implications for near surface geophysical monitoring, Geophysical Research Letters, 34, L18 402, doi: 10.1029/2007GL031124, 2007.

Menke, W.: Geophysical data analysis: discrete inverse theory, Academic Press, doi: 10.2138/am-2017-671, 2012.

Rücker, C. and Günther, T.: The simulation of finite ERT electrodes using the complete electrode model, Geophysics, 76, F227–F238, doi: 10.1190/1.3581356, 2011.

Sen, P. and Goode, P.: Influence of temperature on electrical conductivity on shaly sands, Geophysics, 57, 89–96, doi: 10.1190/1.1443191, 1992.

Telford, W., Geldart, L., and Sheriff, R.: Applied geophysics, vol. 1, Cambridge University Press, Cambride, 1990.

Weigand, M. and Kemna, A.: Multi-frequency electrical impedance tomography as a non-invasive tool to characterize and monitor crop root systems, Biogeosciences, 14, 921–939, doi: 10.5194/bg-14-921-2017, 2017.

Weigand, M. and Kemna, A.: Imaging and functional characterization of crop root systems using spectroscopic electrical impedance measurements, Plant and Soil, doi: 10.1007/s11104-018-3867-3, 2019.

Weigand, M., Flores Orozco, A., and Kemna, A.: Reconstruction quality of SIP parameters in multi-frequency complex resistivity imaging, Near Surface Geophysics, 15, 187–199, doi: 10.3997/1873-0604.2016050, 2017.

Xu, B. and Noel, M.: On the completeness of data sets with multielectrode systems for electrical resistivity survey, Geophysical Prospecting, 41, 791–801, doi: 10.1111/j.1365-2478.1993.tb00885.x, 1993.

Zimmermann, E., Kemna, A., Berwix, J., Glaas, W., and Vereecken, H.: EIT measurement system with high phase accuracy for the imaging of spectral induced polarization properties of soils and sediments, Measurement Science and Technology, 19, 094 010, doi: 10.1088/0957-0233/19/9/094010, 2008.